# Motor actions are spatially organized in motor and dorsal premotor cortex

**Nicholas G Chehade**[1,2,3,4], **Omar A Gharbawie**[1,2,3,4,5]*

[1]Department of Neurobiology, University of Pittsburgh, Pittsburgh, United States; [2]Systems Neuroscience Center, University of Pittsburgh, Pittsburgh, United States; [3]Center for the Neural Basis of Cognition, Pittsburgh, United States; [4]Center for Neuroscience, University of Pittsburgh, Pittsburgh, United States; [5]Department of Bioengineering, University of Pittsburgh, Pittsburgh, United States

**Abstract** Frontal motor areas are central to controlling voluntary movements. In non-human primates, the motor areas contain independent, somatotopic, representations of the forelimb (i.e., motor maps). But are the neural codes for actions spatially organized within those forelimb representations? Addressing this question would provide insight into the poorly understood structure–function relationships of the cortical motor system. Here, we tackle the problem using high-resolution optical imaging and motor mapping in motor (M1) and dorsal premotor (PMd) cortex. Two macaque monkeys performed an instructed reach-to-grasp task while cortical activity was recorded with intrinsic signal optical imaging (ISOI). The spatial extent of activity in M1 and PMd was then quantified in relation to the forelimb motor maps, which we obtained from the same hemisphere with intracortical microstimulation. ISOI showed that task-related activity was concentrated in patches that collectively overlapped <40% of the M1 and PMd forelimb representations. The spatial organization of the patches was consistent across task conditions despite small variations in forelimb use. Nevertheless, the largest condition differences in forelimb use were reflected in the magnitude of cortical activity. Distinct time course profiles from patches in arm zones and patches in hand zones suggest functional differences within the forelimb representations. The results collectively support an organizational framework wherein the forelimb representations contain subzones enriched with neurons tuned for specific actions. Thus, the often-overlooked spatial dimension of neural activity appears to be an important organizing feature of the neural code in frontal motor areas.

*For correspondence:
omar@pitt.edu

Competing interest: The authors declare that no competing interests exist.

## Editor's evaluation

This valuable and technically highly demanding paper combines intra-cortical stimulation and large-field-of view optical imaging to study the forelimb representation in two macaque monkeys. The authors provide convincing evidence that reach-to-grasp and reach-only tasks only activated restricted subset of the forelimb area (as revealed through stimulation). While these results are consistent with the idea of clusters of neural activity that correspond to different forelimb actions, the evidence that this particular claim, as the discussion points out, remains incomplete.

## Introduction

Motor (M1) and dorsal premotor (PMd) cortex in monkeys are widely used as models for studying cortical control of movement. Both cortical areas contain complete motor representations of the forelimb (e.g., *Boudrias et al., 2010*; *Gould et al., 1986*; *Murphy et al., 1978*; *Raos et al., 2003*). Somatotopy of the forelimb representations varies across studies, but there is consensus that muscles and cortical columns do not have a one-to-one mapping. Instead, an arm or a hand muscle receives

corticospinal inputs from a population of cortical columns (*Andersen et al., 1975*; *Rathelot and Strick, 2006*). Similarly, a cortical column can influence activity in several muscles (*Donoghue et al., 1992*; *Fetz and Cheney, 1980*; *Lemon et al., 1987*). The convergence and divergence of corticospinal projections mean that the same motor output is present in many locations within a forelimb representation. For example, intracortical microstimulation (ICMS) evokes shoulder flexion from many M1 sites where the affiliate corticospinal projections target the same group of arm muscles (*Park et al., 2001*). But how can a seemingly simple motor map control a rich repertoire of arm and hand actions?

To address this question, we must first understand the spatial relationship between neural activity that supports arm and hand actions (i.e., function), and the forelimb motor maps (i.e., structure). We consider two different perspectives. The first perspective is that neural codes for arm and hand actions are not spatially organized within the forelimb motor maps. Instead, neural activity affiliated with a particular function (e.g., reach) is present throughout the arm zones, or even throughout entire forelimb representations. This scheme has support in electrophysiological studies that examined relationships between neural coding, recording site location, and forelimb behavior (*Rouse and Schieber, 2016*; *Saleh et al., 2012*; *Vargas-Irwin et al., 2010*). Nevertheless, most electrophysiological investigations pool results across recording sites, offering only limited insight into the spatial organization of functions. An alternative perspective is that neural codes for arm and hand actions are more spatially organized within the forelimb motor maps. Here, neural activity affiliated with a particular action (e.g., reach) is concentrated in subzones within the forelimb representations. This spatial organization of function is consistent with maps obtained using long train ICMS (500 ms), which approximates the duration of natural actions (*Graziano et al., 2002*). The same motor mapping parameters revealed functional subzones in both frontal and parietal cortical areas (*Cooke and Graziano, 2004*; *Gharbawie et al., 2011*; *Graziano et al., 2002*; *Stepniewska et al., 2005*).

Adjudicating between the two perspectives can be facilitated with neuroimaging because it provides uninterrupted spatial sampling and retains the spatial dimension of the recorded activity. The necessary implementation here is to image entire forelimb representations during arm and hand actions then quantify the spatial organization of movement-related cortical activity. Intrinsic signal optical imaging (ISOI), two-photon imaging, and functional magnetic resonance imaging (fMRI) have been successfully used in measuring cortical activity related to arm and hand actions in monkeys (*Ebina et al., 2018*; *Friedman et al., 2020a*; *Kondo et al., 2018*; *Nelissen and Vanduffel, 2011*). The mesoscopic field-of-view (FOV) in ISOI is particularly well suited for the size of the forelimb representations in M1 and PMd. Moreover, ISOI affords high spatial resolution and contrast without extrinsic agents (e.g., GCaMP, dyes, or MION). These reasons motivated us to use ISOI to measure M1 and PMd activity in head-fixed macaques engaged in an instructed reach-to-grasp task (*Figure 1A–D*). In the same FOV, we used ICMS for high-density mapping of the forelimb representations and surrounding territories. We then quantified the spatial overlap between cortical activity determined from ISOI and the forelimb motor map. We reasoned that if the neural code for reaching and grasping is not spatially organized within the forelimb representation, then ISOI should report diffuse task-related activity that overlaps most of the M1 and PMd forelimb representations (*Figure 1B*, top). Alternatively, if the neural code is more spatially organized, then ISOI should report focal task-related activity in subzones of the M1 and PMd forelimb representations (*Figure 1B*, bottom).

## Results

Two monkeys performed an instructed forelimb task that consisted of four conditions: (1) reach-to-grasp with precision grip, (2) reach-to-grasp with power grip, (3) reach-only, and (4) withhold (*Figure 1D*). Both monkeys completed 49 successful trials/condition/session (median; interquartile range [IQR] = 37–51 trials [monkey G], IQR = 45–51 trials [monkey S]). We used ISOI to measure cortical activity during task performance (*Figure 1*). The spatial extent of activity in M1 and PMd cortex was quantified in relation to motor maps derived with ICMS from the same recording chambers. Measurements of joint angles and electromyography (EMG) activity during the task provided context for condition differences in cortical activity.

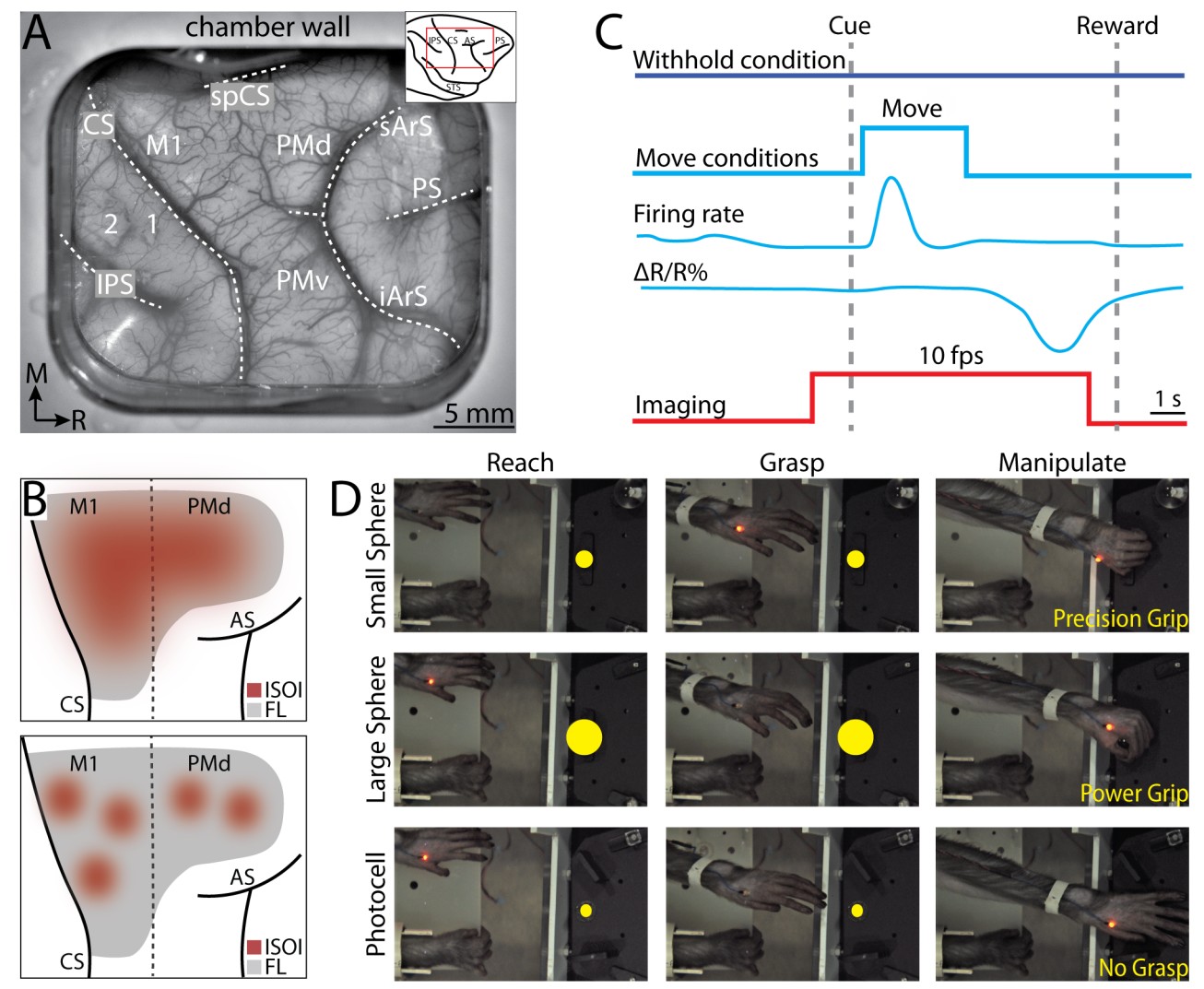

**Figure 1.** Intrinsic signal optical imaging during instructed arm and hand actions. (**A**) Chronic recording chamber provides access to motor and somatosensory cortical areas in the right hemisphere. Native dura is replaced with transparent membrane. Dashed lines mark: central sulcus (CS), intraparietal sulcus (IPS), and arcuate sulcus (AS). Inset shows approximate location of chamber (red rectangle). (**B**) Schematic of potential results. Dotted line separates motor (M1) from premotor areas. Forelimb representation (FL) is gray. Red patches are clusters of pixels that darkened (i.e., negative reflectance) after task-related increase in neural activity. (Top) Pixels reporting activity are in a large patch that overlaps most of FL. (Bottom) Pixels reporting activity are in several small patches that collectively overlap a smaller portion of FL than the patch in A. (**C**) Relative timing in task conditions. Blue square pulse indicates movement period, whereas there was no movement in the withhold condition. Increase in neural firing coincides with movement and precedes reflectance change ($\Delta R/R\%$) measured with intrinsic signal optical imaging (ISOI). Red pulse depicts ISOI acquisition in all conditions (10 frames/s). (**D**) Still frames from three phases (columns) in the three movement conditions (rows). Task was performed with the left forelimb and the right forelimb was restrained. Yellow circles were not visible to the monkey but were digitally added here to facilitate visualization of the three targets.

## Consistent somatotopy in M1 and PMd

To find the forelimb representations in M1 and PMd, we used ICMS to map frontal cortex from central sulcus to arcuate sulcus (*Figure 2A, E*). The general organization of the motor map was consistent between monkeys (*Figure 2B–D, F–H*). Most ICMS sites evoked forelimb movements (*Figure 2B, F*). We marked the rostral border of M1 such that (1) it was 3–5 mm from the central sulcus, and (2) separated high threshold sites (>30 µA) from low threshold sites (*Figure 2C, G*). To simplify the motor map, site classifications were consolidated into broader categories (e.g., elbow and shoulder became arm). The simplified maps (*Figure 2D, H*) showed that the forelimb representations were flanked medially by trunk zones and laterally by upper trunk and face zones. Within the M1 forelimb representation,

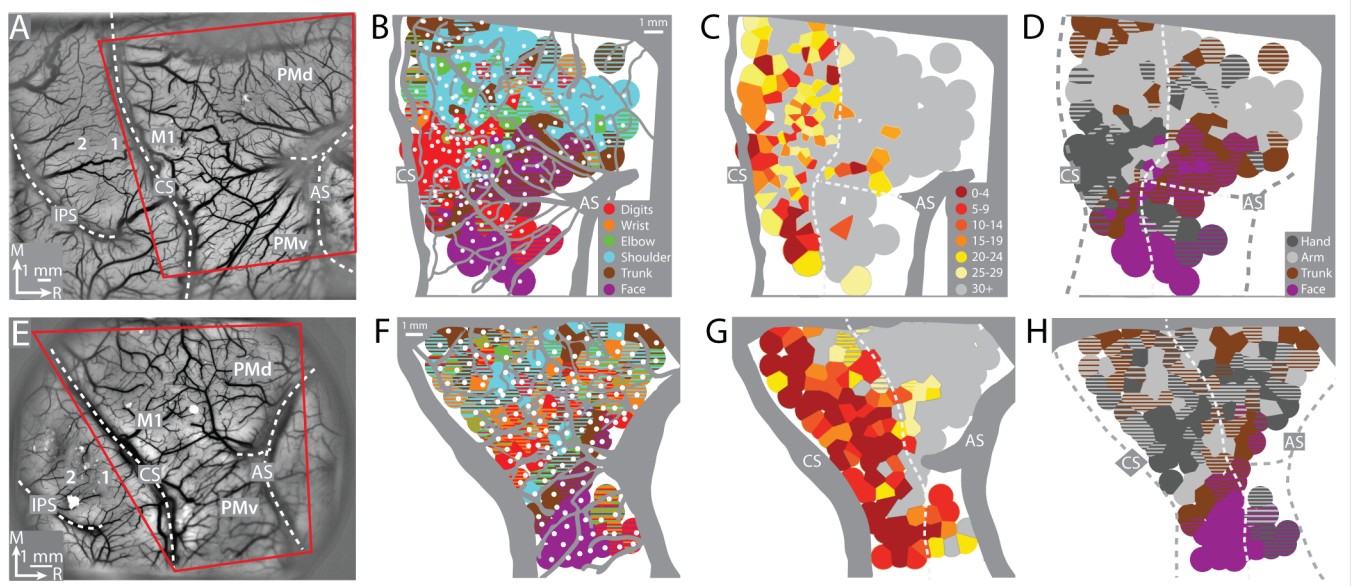

**Figure 2.** Motor map organization in motor (M1) and dorsal premotor (PMd). (**A–D**) Right hemisphere of monkey G. (**A**) Cropped image of a chronically implanted recording chamber. Blood vessels and cortical landmarks are visible through the transparent membrane. Red outline is the field-of-view in subsequent panels and figures. (**B**) Major blood vessels and chamber walls are masked in gray. White dots are intracortical microstimulation (ICMS) sites (n = 218). Voronoi tiles (0.75 mm radius) are color coded according to ICMS-evoked movement. Striped tiles represent dual movements. (**C**) Same motor map from (**B**) colored according to current amplitude (μA) for evoking movements. Border between M1 and premotor cortex is drawn at the transition from low (<30 μA) to high (≥30 μA) current thresholds. (**D**) Same motor map as (**A**), but here wrist and digit sites are classified as hand, and shoulder and elbow sites are classified as arm. (**E–H**) Same as top row, but for right hemisphere of monkey S. Motor map has 158 ICMS sites.

the main hand zone (i.e., digits and wrist) was surrounded by an arm zone, or an arm and trunk zone. This nested organization is consistent with previous maps from macaque monkeys (*Murphy et al., 1978*; *Sessle and Wiesendanger, 1982*) including maps obtained with stimulus triggered averaging of EMG activity (*Park et al., 2001*). The organization of the forelimb representations was less clear in PMd and PMv, which could have been related to higher thresholds, less extensive mapping, as well as more overlap between arm and hand zones than in M1 (*Boudrias et al., 2010*).

## Movement-related activity in M1 and PMd

Neural activity drives a hemodynamic response that is detectable as reflectance change in ISOI. Under red illumination (630 nm wavelength), which was used here, negative reflectance (i.e., pixel darkening) is a lagging indicator of increased neural activity (*Figure 1C*). Thus, in the movement conditions, we assumed that pixels would darken in locations where neural activity increased for movement execution, movement planning, or both. We refer to a spatial cluster of darkened pixels as an active 'patch', which is conceptually similar to an active 'domain', referenced in other studies (e.g., *Bonhoeffer and Grinvald, 1991*; *Lu and Roe, 2007*). A domain, however, is typically identified from the differential cortical response to orthogonal conditions (e.g., orientation domains in V1). If task-related neural activity increases throughout the forelimb representations, then we would expect a large patch to fill most of the FOV (*Figure 1B*, top). If neural activity is more spatially confined, however, then we would expect multiple, smaller, patches (*Figure 1B*, bottom).

First, we examined reflectance change in M1 and PMd in an average time series from a representative session (36 trials/condition). In the power grip condition (*Figure 3A*), there was no reflectance change from baseline to movement onset (−1.0 to +0.5 s from Cue). During reach, grasp, and hold (+0.5 to +2.0 s from Cue), pixels in the center of the FOV brightened (cool colors). In the same period, pixel darkening mostly overlapped the central sulcus and superior arcuate sulcus (landmarks on first panel). At first glance, these observations were counterintuitive as we expected pixel darkening (warm colors) in the center of FOV where neural activity presumably increased in support of arm and hand actions. We will explore the time course of reflectance change in subsequent panels and figures. For now, however, we interpret pixel darkening in this period as indication of activity in major

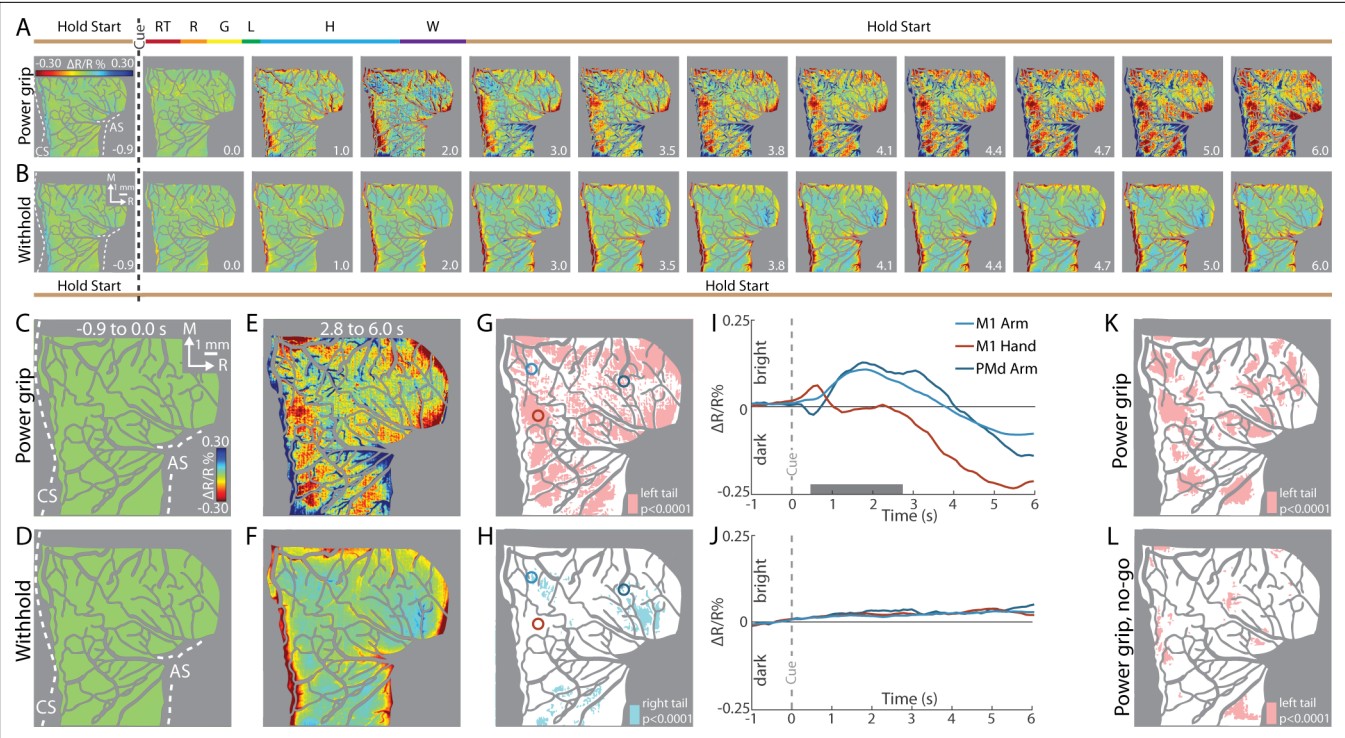

**Figure 3.** Intrinsic signal optical imaging (ISOI) detects movement-related activity in motor (M1) and dorsal premotor (PMd). (**A, B**) Average time series from a representative session (36 trials/condition, monkey G). (**A**) Timeline of average trial from the power grip condition. Hold start = hand in start position. Other phases include reaction time (RT), reach (R), grasp (G), lift (L), hold (H), and withdraw (W). Select frames with time (seconds from cue onset) in bottom right. Reflectance change was clipped to the median pixel value ±0.3 standard deviation (SD). Clusters of pixels started to darken (hot colors) at 3.5 s and gradually increased in size and intensity. (**B**) Matching frames from the withhold condition. Color scale is same as (**A**). (**C, D**) Baseline frames in the power grip and withhold conditions. (**E**) Movement frame in the power grip condition: mean of 33 frames captured from movement completion until end of trial. (**F**) Temporally matched mean frame from the withhold condition. (**G**) Thresholded map from the power grip condition. Red pixels were significantly darker (*t*-test, left tail, p < 0.0001) in (**E**) than in (**C**). Colored circles (0.40 mm radius) are regions-of-interest (ROIs) placed in M1 hand, M1 arm, and PMd arm. (**H**) Thresholded map from the withhold condition. Cyan pixels were brighter (*t*-test, right tail, p < 0.0001) in (**F**) than in (**D**). (**I**) Time courses of reflectance change in the power grip condition. Line colors match the ROIs. Negative values indicate pixel darkening. Gray horizontal bar depicts mean movement duration. (**J**) Same as (**I**), but for the withhold condition. (**K**) Thresholded map from the power grip condition (2 sessions, 159 trials). Red pixels darkened after movement as compared to baseline (*t*-test, left tail, p < 0.0001). (**L**) Thresholded map from the no-go condition (2 sessions, 159 trials). Only a small number of pixels darkened in M1 and PMd.

The online version of this article includes the following figure supplement(s) for figure 3:

**Figure supplement 1.** Movement observation drives reflectance change in dorsal premotor (PMd) but not in motor (M1).

vessels. Once the hand returned to the start position after movement completion (i.e., hold start), we observed the expected reflectance change: pixels in the center of the FOV started to darken and form patches in M1 and PMd (+3.5 s from Cue). Patches continued to darken and expand in the remaining frames. Nevertheless, even at peak intensity and size (+5.0 to +6.0 s from Cue), patches remained spatially separable within M1 and between M1 and PMd. For contrast, *Figure 3B* shows an average time series from the withhold condition where the Cue instructed the animal to hold the start position for the duration of the trial. Here, pixel darkening was limited to major blood vessels as most of the FOV was unchanged or brightened (*Figure 3B*, cyan pixels). Thus, the patches in *Figure 3A* were likely lagging indicators of movement-related neural activity.

To summarize the activity patterns in the time series, we averaged frames from two phases. For the baseline phase, we averaged 10 frames acquired −1.0 to 0 s from Cue. For the movement phase, we averaged 33 frames captured post-movement (+2.8 to +6.0 s from Cue). Average baseline frames showed no reflectance change and were indistinguishable between the movement and withhold conditions (*Figure 3C, D*). In contrast, reflectance change was apparent in the movement frame (*Figure 3E*), but not in the matching period from the withhold condition (*Figure 3F*). To threshold *Figure 3E*, we flagged pixels that darkened significantly in the post-movement frame as compared

to the baseline frame (*Figure 3G*, *t*-test, left tail, p < 0.0001). Thus, red pixels in *Figure 3G* show locations of movement-related neural activity from a single session. We thresholded *Figure 3F* for contrast, but to capture the predominant reflectance change here, we focused on pixels that brightened in the temporally matched frame as compared to the baseline frame (*Figure 3H*, *t*-test, right tail, p < 0.0001). The small number of zones and their small size indicates that reflectance change was limited in the absence of movement.

Next, we examined the time course of reflectance change in both conditions (*Figure 3I, J*). We placed regions-of-interest (ROIs) with two guiding objectives (*Figure 3G*, colored circles). First, to target M1 hand, M1 arm, and PMd arm, which we achieved by consulting the motor maps and blood vessel patterns. Second, to target locations that showed activity in time series averaged across trials from all imaging sessions (e.g., Figure 5A). Time courses from two of the ROIs showed an abrupt, yet small, reflectance change with movement onset (+0.5 s from Cue). This brief change was likely due to motion artifact and will be more apparent in the next figures. After the artifact, the arm ROIs and the hand ROI had distinct time course profiles (*Figure 3I*). For the arm ROIs, positive reflectance started to increase ~0.3 s after movement onset (+0.8 s from Cue) and peaked during the hold phase (+1.7 s from Cue) when the object was grasped and maintained in the lifted position. Then, negative reflectance increased and peaked 2–3 s after movement completion (+5.0 to +6.0 s from Cue). The positive–negative sequence is discussed later in relation to triphasic time courses (negative–positive–negative) established for sensory cortex (*Chen-Bee et al., 2007*; *Sirotin and Das, 2009*). In contrast, the time course of the hand ROI did not show the positive reflectance that lasted for several seconds in the arm ROIs. Instead, the time course was flat until negative reflectance began to increase +2.5 s from Cue. The timing of the negative peak was consistent, however, across the hand and arm ROIs. The distinctiveness of the time courses from the arm and hand may reflect functional differences between those zones. In contrast to the profiles in *Figure 3I*, the same ROIs showed no reflectance change in the withhold condition (*Figure 3J*).

## Pixel darkening in M1 and PMd is locked to movement

Our next objective was to determine whether movement *execution* was necessary for the pixel darkening in the movement condition. We addressed this point in three separate experiments. In *Experiment 1*, one monkey performed reach-to-grasp trials interleaved with no-go trials. In a no-go trial, the Go Cue and preceding steps were no different from a trial in a movement condition. In a no-go trial, however, 250 ms after the Go Cue (i.e., during reaction time), another Cue (LED plus tone) instructed the monkey to hold the start position instead of move. Correct trials were typically achieved with the monkey releasing the start position to reach and then immediately returning its hand to the start position. Movements in no-go trials were therefore truncated but not inhibited. If the monkey reached past the midpoint between the start position and the target, then the trial was considered incorrect. We reasoned that no-go trials would minimize movement-related cortical activity without interfering with internal processes that precede movement (e.g., movement preparation). Results were obtained from two imaging sessions. In the movement condition, the thresholded map showed patches of activity in post-movement frames as compared to baseline frames (*Figure 3K*, *t*-test, p < 0.0001). In contrast, the thresholded map from the no-go condition had fewer and smaller patches (*Figure 3L*). *Experiment 1* therefore shows that significant pixel darkening was contingent on movement execution.

In *Experiment 2*, we added conditions in which the monkey did not move but observed an experimenter perform the task instead. First, the monkey completed blocks of trials with regular movement conditions (performed trials). Next, the monkey observed the experimenter perform blocks of trials of the same movement conditions (observed trials). The primate chair was closed off in the observed trials to prevent the monkey from reaching. Cues, timing, and reward schedule were consistent between performed and observed trials. Time courses were measured from ROIs in M1 and PMd (*Figure 3—figure supplement 1A*). In M1, time courses had a clear negative peak in the performed trials (*Figure 3—figure supplement 1B*) but remained near baseline in the observed trials. In PMd however, the performed and observed trials had overlapping time courses. All negative peaks in PMd were smaller than the M1 peaks in the performed trials. Results from *Experiment 2* suggest that M1 activity was driven by movement execution, whereas PMd activity was likely related to movement execution, movement preparation, and motor cognition.

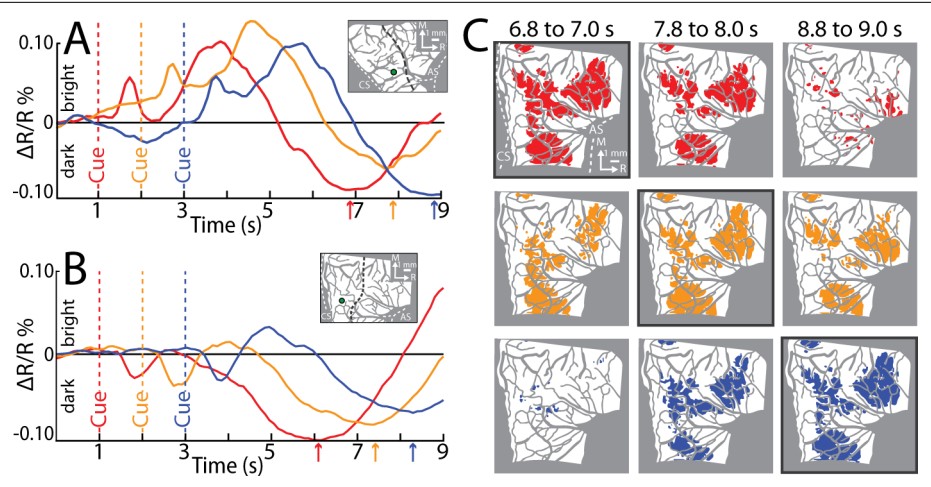

**Figure 4.** Reflectance change is locked to movement. (**A**) Time course of reflectance change in motor (M1). Average of 2 sessions (95 trials/condition, monkey S). Inset shows region-of-interest (ROI; 0.40 mm radius). Line plot colors match cue condition (dashed line). Arrows point to negative peaks. (**B**) Same as (**A**), but for monkey G (1 session, 44 trials/condition). (**C**) Thresholded maps from the data in (**B**). Each map is an average of 3 frames and reports activity from different time points in relation to Cue. Colored pixels darkened in the movement frame as compared to baseline (*t*-test, p < 0.0001). Pixel colors match cue condition (1 condition/row). A black border is drawn around the thresholded map that is +5.8 to +6.0 s from Cue.

In *Experiment 3*, we systematically varied the timing of the Go Cue so that it was 1, 2, or 3 s from trial initiation. We reasoned that temporal shifts in movement onset would lead to predictable shifts in pixel darkening. Only one movement condition was tested in this experiment. We placed ROIs in the M1 hand zone (insets in *Figure 4A, B*). Time courses from both monkeys confirmed that the negative peaks were locked to Cue onset, and by extension movement onset. This temporal relationship was also evident in the spatial development of cortical activity (*Figure 4C*). For each Cue condition, we generated thresholded maps from three time windows near the negative peaks. Each thresholded map therefore reported on pixels that darkened in a specific time window as compared to the baseline frame (*Figure 4*, *t*-test, p < 0.0001, *n* = 44 trials). Across Cue conditions, the largest maps were in the time window +5.8 to +6.0 s from Cue (*Figure 4C*, outlined panels). Thus, *Experiment 3* shows that peak map size, and the timing of the negative peak, were both locked to movement onset. Collectively, *Experiments 1–3* confirm that the M1 activity reported here was linked to movement execution, whereas PMd activity was likely more complex.

## Average time series reveal consistent features of M1 and PMd activity

To identify the most consistent activity patterns, we generated an average time series for each condition. *Figure 5A* summarizes the average time series for the precision grip condition (monkey G, 8 sessions, 404 trials). Animating the time series side-by-side with a representative behavioral trial (*Video 1*) qualitatively revealed spatiotemporal features of the cortical activity. (1) During movement, pixel darkening was limited to major vessels; the rest of the FOV brightened or was unchanged. (2) Pixel darkening in cortex (i.e., beyond major vessels) occurred after movement completion. (3) The intensity of pixel darkening and the number of pixels that darkened progressed gradually from movement completion until the end of image acquisition. The same features were present in the other movement conditions (not shown).

From the average time series, we condensed the spatiotemporal organization of cortical activity into a single frame. To that end, we used the time course of every pixel to visualize the time of the negative peak (i.e., maximal darkening). *Figure 5B* color codes peak times from the time series in *Figure 5A* but omits information related to magnitude of reflectance. Peak times fit into three broad categories (*Figure 5B*). (1) *Early peaks*. Green pixels peaked ~2 s from Cue when the monkey was still engaged in task-related movements. Green pixels generally corresponded with the hot color pixels in *Figure 5A* panels 1.6–2.5, and therefore may have been affiliated with the early response in major

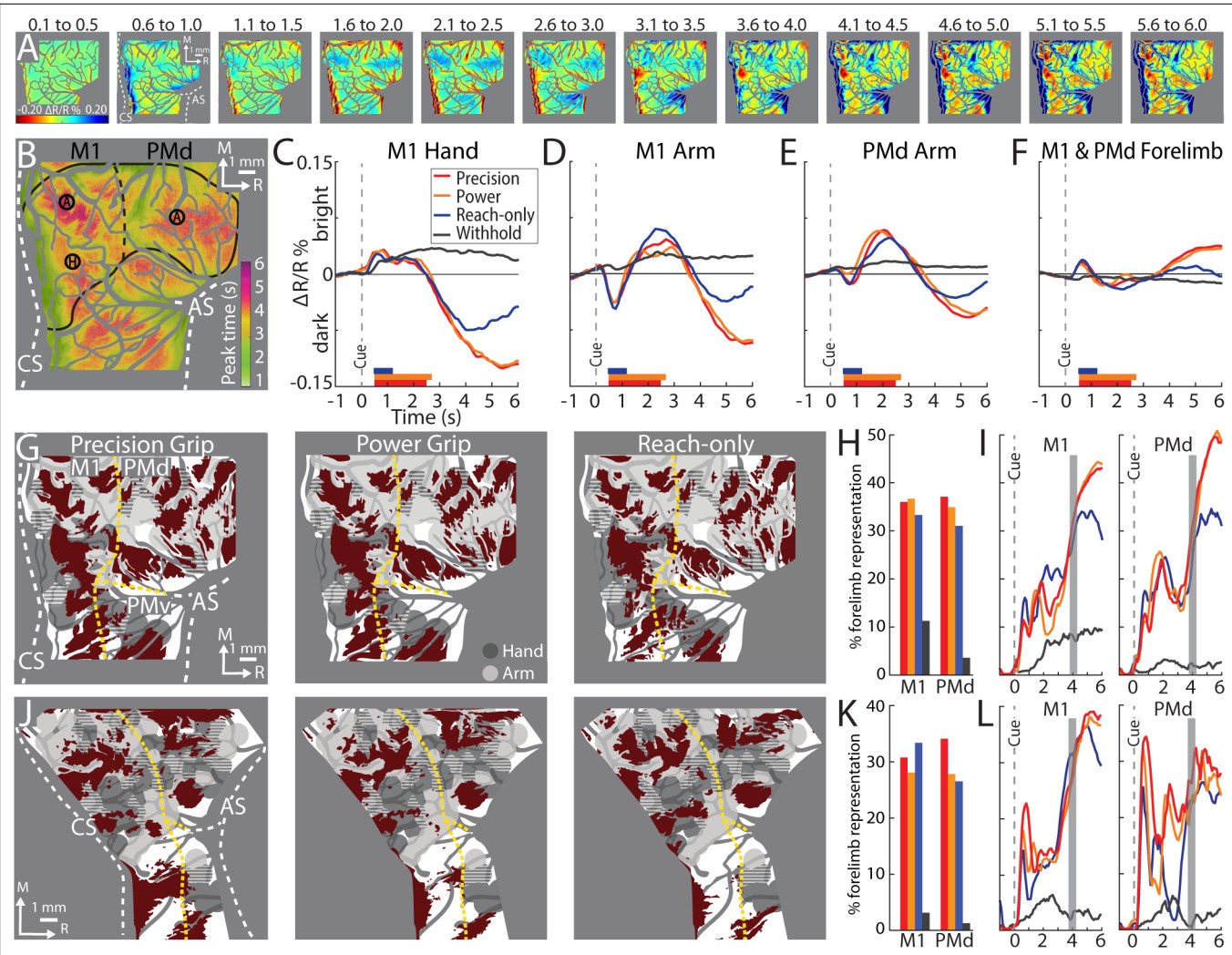

**Figure 5.** Average time series reveal consistent features of cortical activity. (**A**) Average time series for precision grip (8 sessions, 404 trials, monkey G). Each frame is the average of five successive frames, averaged across trials. Time above each frame is seconds from cue onset. Reflectance change was clipped to the [median ± standard deviation (SD)] pixel value across frames. Patches started to form at 3.1–3.5 s. (**B**) Full field-of-view (FOV) colored according to the time (seconds from Cue) that pixels reached maximal negative reflectance. Hot colored pixels peaked relatively late. Black circles (0.40 mm radius) in motor (M1) and dorsal premotor (PMd) are regions-of-interest (ROIs) placed in arm "A" and hand "H zones. The forelimb representations are outlined in black with the dotted line separating M1 from PMd. The entire outline is also used an ROI. (**C–F**) Each plot is based on one of the ROIs in (**B**). Horizontal bars on x-axis show the average movement duration of each condition. (**G**) Thresholded maps for each condition superimposed on the forelimb motor map. Red pixels darkened in the post-movement frame (5-frame average) as compared to the baseline frame (t-test, p < 0.0001). Dashed yellow lines mark cortical borders. (**H**) Percentage of forelimb representations with red pixels from (**G**). Bar colors match (**C**). (**I**) Same quantification in (**H**) expressed as a function of trial duration. Thresholded maps were generated at every time point (0.1 s) with a t-test (p < 0.0001) comparison of the frame at the time point and the baseline frame. (**J–L**) Same as (**G–I**), but for monkey S.

The online version of this article includes the following figure supplement(s) for figure 5:

**Figure supplement 1.** Thresholded maps corrected for multiple comparisons overlap small parts of the forelimb representations.

**Figure supplement 2.** Thresholded maps from all post-movement frames overlap small parts of the forelimb representations.

vessels. (2) *Intermediate peaks*. Orange/yellow pixels peaked ~3.5 s from Cue, which was within 1 s of movement completion. Orange/yellow pixels were in the approximate location of yellow pixels in *Figure 5A* last panel. (3) *Late peaks*. Magenta/red pixels peaked ~5.5 s from Cue, which was ~3.0 s from movement completion. These pixels corresponded with the hot color pixels in *Figure 5A* last three panels. Late peak pixels were in clusters surrounded by intermediate peak pixels, but there was no regional divide between the two.

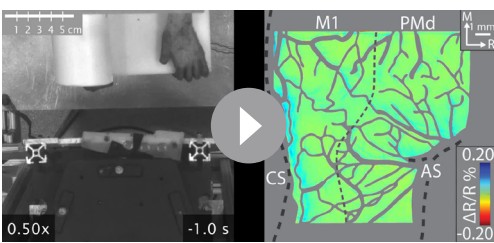

**Video 1.** Pixel darkening in intrinsic signal optical imaging (ISOI) lags movement. *Left*: Representative trial from the precision grip condition filmed from overhead (monkey G). The task was performed with left forelimb and the right forelimb was restrained. Video plays at half-speed. Time in bottom right is relative to start of ISOI acquisition; Cue onset was at 1.0 s. Movement started at 1.3 s and was complete by 3.6 s. *Right*: Average ISOI time series (8 sessions, 404 trials) from the same condition. In addition to the image processing described in the methods, frames were temporally smoothed with a 5-frame sliding window. Note, that patches did not form until movement completion.
https://elifesciences.org/articles/83196/figures#video1

For another perspective on the spatiotemporal patterns of cortical activity, we examined time courses from ROIs that we placed in M1 and PMd (*Figure 5B*). The three circular ROIs matched those in *Figure 3G* and returned time courses consistent with *Figure 3I, J*. In the movement conditions, time courses from the three ROIs had motion artifact that started with reach onset (*Figure 5C–E*; ~0.5 s from Cue) and lasted for ~0.2 s. For the M1 hand ROI (*Figure 5C*), after the artifact, reflectance change was flat for ~1.5 s. Negative reflectance started to increase ~2.5 s from Cue, which coincided with the end of movement in the precision and power conditions. In those conditions, the negative peak was larger, and occurred later, than in the reach-only condition. This temporal difference could have been due to the larger size of the negative peaks, or longer movement durations, or both. ROIs in M1 and PMd arm zones returned time courses that were consistent with one another (*Figure 5D, E*). In both ROIs, the motion artifact was followed by an increase in positive reflectance, which is quite different from the time courses from the hand ROI (*Figure 5C*). Nevertheless, in the arm ROIs, negative reflectance increased from +3.5 from Cue and peaked at approximately the same time as the M1 hand ROI.

Two controls provided context for the time courses of the movement conditions (*Figure 5C–E*). First, in all ROIs, the withhold condition had time courses that remained close to baseline (*Figure 5C–E*). This control confirms that the reflectance change in the movement conditions was movement driven. Second, a ROI that comprised the entire forelimb representation of M1 and PMd (*Figure 5B*) returned time courses that differed entirely from the other three ROIs. Most importantly, the forelimb ROI lacked the characteristic negative peaks in *Figure 5C–E*. This control confirms that time courses from movement conditions were spatially specific responses that did not generalize to the entire forelimb representation.

## Thresholded activity maps overlap limited portions of the forelimb representations

Next, we generated thresholded maps to summarize the spatial patterns of movement-related cortical activity. For each condition, trials were pooled across sessions and every trial contributed a baseline frame and a movement frame. Pixels that darkened significantly (*t*-test) in the movement frame as compared to the baseline frame were included in the thresholded map. A baseline frame was the average of the first 10 frames in a trial. The movement frame was defined in two different ways. First, as an average of all frames (*n* = 39) captured from end of movement (+2.2 to +6.0 s from Cue). This strategy was adopted for the thresholded maps in *Figure 3*. Second, as an average of 5 consecutive frames captured in the post-movement period. A similar strategy was adopted in *Figure 4*, but here the 5-frame range was guided by the timing of the negative peaks. Thus, we averaged the values in *Figure 5B* to determine the mean timing of negative peaks in the precision condition. We repeated the procedure for the other conditions and then averaged the times across conditions. From those averages, the 5 frames for calculating the movement frame were set to +4.1 to +4.4 s from Cue (monkey G) and +3.9 to +4.3 s from Cue (monkey S).

First, we examine the 5-frame thresholded maps (*t*-test, p < 0.0001). We co-registered those maps with the motor maps for reference (*Figure 5G, J*). The general organization of the thresholded maps was more similar across conditions within an animal than for the same condition across animals (*Figure 5G, J*), which is consistent with human fMRI maps of M1 finger zones (*Ejaz et al., 2015*). In all thresholded maps, significant pixels were organized in patches that overlapped subzones of the M1

and PMd forelimb representations. The most lateral patches were in M1 face and PMv forelimb and were therefore excluded from subsequent analyses. In each map the patches collectively overlapped only 31% (median, IQR = 29–35%) of the forelimb representations (*Figure 5H, K*). This measurement takes into consideration both monkeys, both M1 and PMd, and the three movement conditions. For reference, thresholded maps from the withhold condition overlapped 3% (median, IQR = 1–7%) of the M1 and PMd forelimb representations.

We repeated the overlap measurements after re-thresholding the maps to correct for multiple comparisons based on the number of pixels in the forelimb representations (*t*-test, p < 1e−7). This more conservative threshold did not impact the spatial organization of the maps, but it shrank their sizes to 14% (median, IQR = 16–20%) overlap with the forelimb representations (*Figure 5—figure supplement 1*). In contrast, expanding the frame range of thresholded maps to the 39 frames post-movement (*t*-test, p < 0.0001) led to only a small reduction of overlap with the forelimb representations (*Figure 5—figure supplement 2*; median = 27%, IQR = 25–34%). Thus, generating thresholded maps using a less conservative threshold, or using different frame ranges from the post-movement period, returned the same overall result: <40% overlap with the M1 and PMd forelimb representations.

For context on map sizes reported thus far, we redid the overlap measurements at every time point. Thus, we generated a thresholded map (*t*-test, p < 0.0001) for every frame and measured its overlap with the M1 and PMd forelimb representations (*Figure 5I, L*). A consistent feature across movement conditions, cortical areas, and monkeys, was that the size of thresholded maps started to rapidly increase from ~3 s after Cue, which nearly coincided with the end of movement in the precision and power conditions. The size of the maps plateaued, or even decreased, by ~5 s from Cue. At peak map

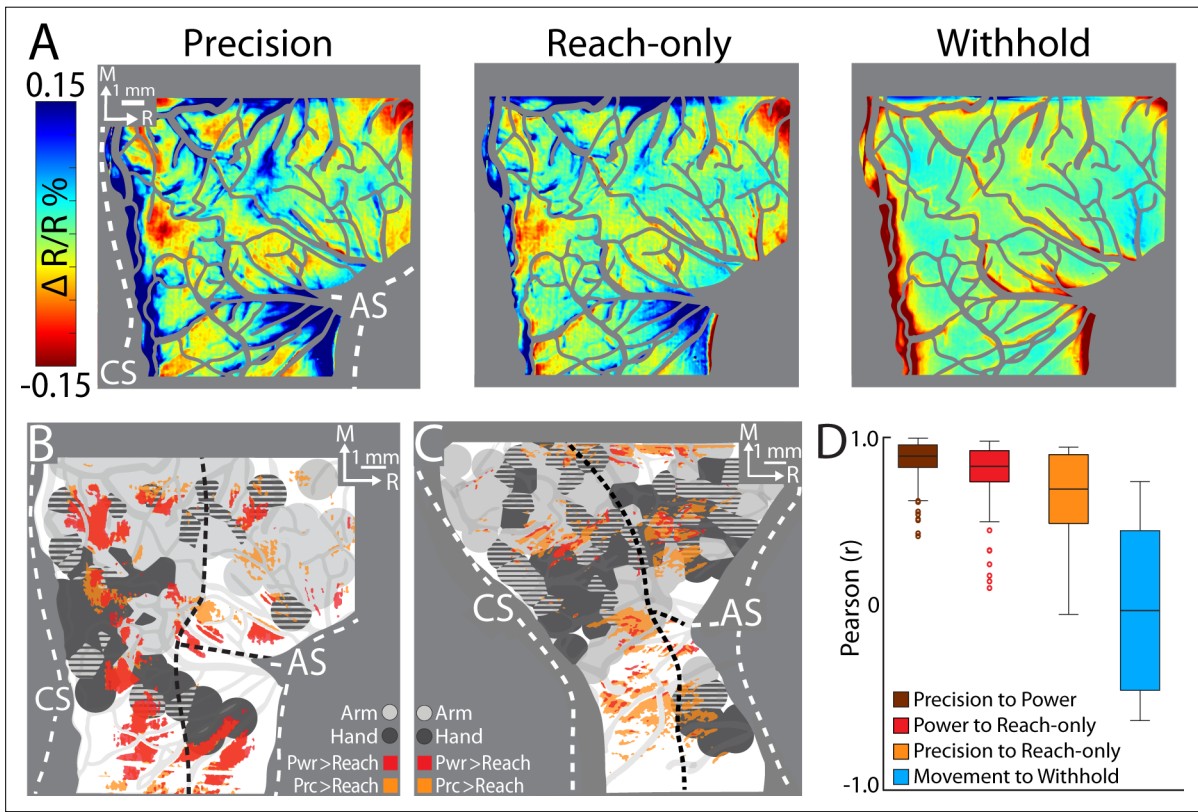

**Figure 6.** Comparing cortical activity between conditions. (**A**) Movement frames (+2.2 to +6.0 s from Cue) averaged across conditions (*n* = 326–358 trials, monkey G). Reflectance change was clipped to the median pixel value ±1.5 standard deviation (SD) among all conditions. (**B**) Thresholded maps from two paired-comparisons. Red and orange pixels flag locations that darkened significantly in the power (pwr) and precision (prc) conditions, respectively, over the reach-only (*t*-test, p < 0.001). Dashed black lines mark cortical area borders. (**C**) Same as (**B**) for monkey S. (**D**) Cross-correlation distributions of trial averaged frames for conditions pairs. A data point here is the coefficient from cross-correlating (zero lag) one trial averaged frame of a condition with the time-matched frame of another condition. Each box, therefore, included 78 points (39 frames × 2 monkeys). The top and bottom of each box are the first and third quartiles of the data; whiskers span 1.5× the interquartile range. The blue box includes 234 points from comparing three movement conditions to the withhold condition (39 frames × 3 comparisons × 2 monkeys).

expansion, overlap with the forelimb representations was 38% (median, IQR = 35–44%). Thresholded maps in (*Figure 5* and *Figure 5—figure supplement 2*) were therefore only slightly smaller than peak maps sizes. Thus, the two strategies that we adopted for selecting movement frames (5- and 39-frame averages) did not underreport the size of the thresholded maps.

## Condition differences in cortical activity patterns

The spatial organization and size of the thresholded maps suggested similar cortical activity across conditions. Nevertheless, time courses (*Figure 5C–E*) and non-binarized average maps (*Figure 6A*) showed greater magnitudes of activity in the reach-to-grasp conditions as compared to the reach-only condition. This motivated us to directly compare cortical activity between conditions, which we achieved in two ways.

First, we conducted *t*-test comparisons on movement frames (39-frame average) from condition pairs (*Figure 6B, C*). Our objective was to generate thresholded maps of pixels that darkened significantly in one movement condition relative to another movement condition. Maps thresholded at p < 0.0001 did not reveal distinguishable patches. After relaxing the threshold to p < 0.001, we found several patches in the M1 and PMd forelimb representations that darkened more in the reach-to-grasp conditions as compared to the reach-only condition (*Figure 6B, C*). In contrast, fewer and smaller patches reported differences between the precision and power conditions (not shown). Thus, the thresholded maps in *Figure 6B, C* flag cortical zones that were more active in the reach-to-grasp conditions as compared to the reach-only condition.

Second, we used cross-correlation at zero lag to compare average time series from pairs of conditions. We focused on the post-movement frames and correlated each frame from one condition with

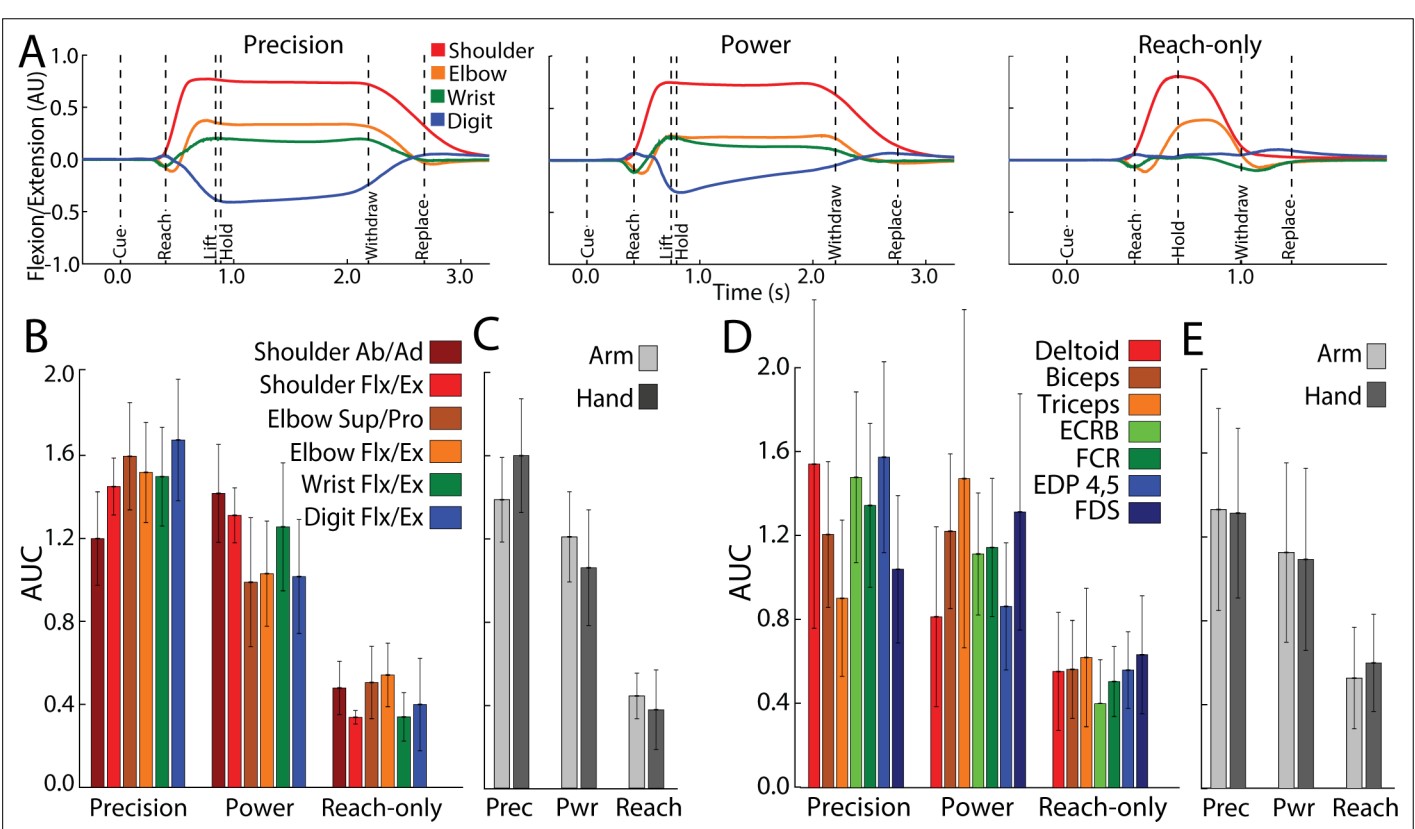

**Figure 7.** Forelimb activity scales across conditions. (**A**) Mean flexion/extension changes in two arm joints and two hand joints as a function of time in the three movement conditions (5–15 sessions, 143–405 trials, monkey G). *Y*-axis scale is arbitrary units (AU). (**B**) Area under the curve (AUC, mean ± standard deviation [SD]) for joint activity from trials in (**A**). (**C**) Pooled AUC (mean ± SD) for joint activity in the three conditions. Shoulder and elbow joints were classified as arm. Wrist and digit joints were classified as hand. (**D, E**) Same as (**B, C**), but for muscle activity (6–14 sessions, 308–595 trials, monkey G). Deltoid, biceps and triceps were classified as arm. ECRB (extensor carpi radialis brevis), EDP (extensor digitorum profondus), FCR (flexor carpi radialis) and FDS (flexor digitorum superficialis) were cassified as hand.

the time-matched frame in the paired condition. The analysis was limited to pixels within the fore-limb representations. Correlation coefficients were pooled across monkeys such that each condition pair returned a distribution of 78 points (39 frames × 2 monkeys). A one-way analysis of variance (ANOVA) showed an effect of condition pairs on the correlation coefficients ($F_{(2,231)} = 12.98$, $p < 0.01$; *Figure 7D*). Post hoc tests corrected for multiple comparison ($n = 3$; $p < 0.01$), showed that the correlation coefficients for *precision X reach-only* was lower than the other two condition pairs. The cross-correlations therefore show that the spatiotemporal patterns of cortical activity were most dissimilar between the precision and reach-only conditions. For context, cross-correlating movement conditions with the withhold condition returned a much lower distribution of coefficients (*Figure 6D*). Cross-correlation was therefore sensitive enough to detect condition differences in the time series. In sum, the two approaches that we used for directly comparing cortical activity between conditions indicate that condition differences were small but most pronounced between the precision and reach-only conditions.

### Forelimb use differences across conditions

We examined whether forelimb use could shed light on the cortical activity patterns observed in the movement conditions. From the consistent sizes of thresholded maps (*Figure 5H, K*), we expected similar forelimb use across conditions. In contrast, from the negative peak magnitudes in *Figure 5C–E* and the condition differences in *Figure 6B–D*, we expected greater forelimb activity in the reach-to-grasp conditions as compared to the reach-only condition. To test these possibilities, we measured forelimb kinematics for 12 degrees of freedom and measured EMG activity from seven forelimb muscles.

*Figure 7A* shows average time courses for 8 degrees of freedom in arm and hand joints. Shoulder and elbow time courses were similar in the precision and power conditions. In the precision condition, however, digit and wrist angles were sustained throughout the hold period. The reach-only condition differed from the other two conditions in having truncated time courses and nearly absent flexion/extension of the digits. To quantify condition differences, we measured the area under the curve (AUC) from Cue to Replace. AUC values were standardized for each joint to facilitate comparisons across conditions. The reach-only condition stood out for having the smallest AUC (*Figure 7B*) in all degrees of freedom. To simplify condition comparisons (*Figure 7C*), we first combined the AUC calculated for arm joints (shoulder and elbow) and separately combined the AUC for hand joints (wrist and digits). For statistical testing, we then combined the AUC for arm and hand into a single group. A one-way ANOVA detected a main effect of movement condition for AUC values of forelimb kinematic activity ($F_{(2,4332)} = 6194$, $p < 0.0001$). Post hoc tests corrected for multiple comparisons ($n = 3$; $p < 0.01$) indicated that the kinematics of forelimb joints was less for the reach-only condition than the two grasp conditions. Furthermore, the kinematics of forelimb joints was smaller for the power condition than for the precision condition.

The same condition differences in joint activity were evident in the activity of individual arm and hand muscles (*Figure 7B*). We combined the muscle AUC measurements into groups as was done for the joints. We found the same main effects and post hoc condition differences in the muscle AUC measurements [*Figure 7D*; ANOVA, $F_{(2,9794)} = 3633$, $p < 0.0001$]. Thus, joint angle kinematics and EMG indicate that arm and hand activity systematically differed across conditions: precision > power > reach-only. These forelimb use differences across conditions are consistent with the dissimilarities in cortical activity between the precision and reach-only conditions (*Figure 5C–E* and *Figure 6B–D*).

## Discussion

We investigated the relationship between movement-related cortical activity and motor maps in M1 and PMd. ISOI showed that cortical activity that supports reaching and grasping is concentrated in patches within the forelimb representations. The spatial configuration of the patches was consistent across task conditions despite variations in the reach-to-grasp movements. Nevertheless, the magnitude of activity within the patches scaled with forelimb use as measured from joint and muscle activity. The spatial organization of thresholded maps, and relative invariance across conditions, suggests that neural activity for arm and hand functions is not uniformly distributed throughout the forelimb

representations. Instead, our observations are consistent with a functional organization wherein subzones of the forelimb representations are preferentially tuned for certain actions.

## Measuring activity in M1 and PMd

Two provisions facilitated our findings. First, we used ISOI to measure cortical activity. ISOI is well established for studying sensory cortex (*Bonhoeffer and Grinvald, 1991*; *Chen et al., 2001*; *Grinvald et al., 1986*; *Ts'o et al., 1990*), but few studies have leveraged it for cortical control of movement (*Friedman et al., 2020a*; *Siegel et al., 2007*). Functional MRI has been used more extensively to investigate reaching and grasping in monkeys and humans (e.g., *Cavina-Pratesi et al., 2010*; *Gallivan and Culham, 2015*; *Nelissen and Vanduffel, 2011*). Our objective, however, was better served with the higher contrast and spatial resolution afforded from ISOI. Similarly, calcium imaging has gained traction in cortical control of movement and brain–computer paradigms because it provides single-cell resolution and better temporal fidelity to neural signals. But in monkeys, the FOV in mini-scopes and two-photon microscopes, covers only a small fraction of one forelimb representation (*Bollimunta et al., 2021*; *Ebina et al., 2018*; *Kondo et al., 2018*; *Trautmann et al., 2021*). In contrast, the present FOV was large enough for all forleimb representations on the precentral gyrus (M1, PMd, and PMv). Second, we obtained motor maps from the same chronic chambers and used them to quantify the spatial extent of the cortical activity recorded with ISOI. Without the present high-density motor maps, it would not have been possible to distinguish between cortical activity in the forelimb representations and the counterpart in the surrounding trunk and face zones. Thus, conducting ISOI and motor mapping in the same hemispheres was central to our results and interpretations.

Nevertheless, our approach had limitations that must be considered for proper interpretation of the results. First, our monkeys performed a reaching task that explored only a narrow range of their repertoire of arm and hand actions. Diversifying the task conditions would have expanded the range of arm and hand movements, which could have provided additional insight into the organization of cortical activity that supports forelimb actions. Second, the motor maps were based on low dimensional classifications of the ICMS-evoked movements. Although our motor mapping approach was sufficient for distinguishing the forelimb representations from the surrounding zones, it may have oversimplified the spatial organization of the arm and hand zones relative to each other. For that reason, we measured the spatial extent of the cortical activity with respect to the forelimb representations as a whole and did not examine potential differences between arm and hand zones.

## Delayed negative peaks are locked to movement

Pixel darkening peaked several seconds after movement onset. Our three control experiments and the withhold condition indicated that most of the present pixel darkening was predicated on movement execution. The reflectance change time courses were consistent with those reported in the few studies that used ISOI with forelimb tasks (*Friedman et al., 2020a*; *Heider et al., 2010*). Nevertheless, the lag between movement onset and peak pixel darkening was two to three times longer than expected from stimulus onset (peripheral or ICMS) and peak darkening in sensorimotor cortex (e.g., *Card and Gharbawie, 2020*; *Chen et al., 2001*; *Friedman et al., 2020b*). Slower intrinsic signals have been reported in other behavioral paradigms (*Grinvald et al., 1991*; *Tanigawa et al., 2010*), which raises the possibility of time course differences between awake and anesthetized ISOI. Fundamental differences could also exist between intrinsic signals evoked from movement (active) and those from stimulation (passive). In our task, the arm and hand were active for ~2 s per trial. In contrast, sensory stimuli are typically more confined in both space and time to optimize focal activation in cortex (e.g., S1 barrel, V1 orientation column).

The protracted peak darkening should be considered in relation to the triphasic time course of reflectance change in ISOI (*Chen-Bee et al., 2007*; *Sirotin et al., 2009*). Studies with stimulus-evoked responses typically focus on the *initial dip*, which is the first dark peak and occurs 1–2 s after stimulus onset. The subsequent *rebound* (bright peak) and *undershoot* (second dark peak) unfold sequentially over several seconds. In our time courses, the timing of the darkest peak would have coincided with the *rebound* in stimulus-evoked responses. The timing of our dark peak can potentially fit with the triphasic response in three ways. (1) Our dark peaks could have been *initial dips* locked to movement onset. In this interpretation our movement-evoked time courses would be slower iterations of stimulus-evoked responses. (2) Our dark peaks could have been *initial dips* locked to movement

offset. The early negative peak of the reach-only condition does not provide clear insight into this point because this condition had an early offset time as well as a relatively small negative peak that could have peaked early. Direct testing of the movement offset interpretation would therefore require systematic manipulation of movement duration, which we did not explore. (3) Our dark peaks could have been the *undershoot* locked to movement onset. In this case, the *initial dip* may have been undetected because it was too small, obscured by motion artifact, or both. In this interpretation our movement-evoked time courses would be consistent with stimulus-evoked responses, albeit with a relatively brief rebound.

## Movement activates subzones of the forelimb representations

Average thresholded maps reported cortical locations where task-related activity was present in hundreds of trials. Those maps overlapped <40% of the M1 and PMd forelimb representations. Expanding the number of post-movement frames had no impact on the size of the thresholded maps and correcting the maps for multiple comparisons further diminished their size. Relatively small size was therefore a robust feature of the thresholded maps and we consider potential explanations for that result. (1) *ISOI sensitivity*. It is unlikely that ISOI lacked sensitivity for capturing the full extent of cortical activity and thereby underreported thresholded map sizes. If anything, ISOI sensitivity to subthreshold electrophysiological signals can lead to overestimation of the spatial extent of modulated neural activity (*Frostig et al., 2017*; *Grinvald et al., 1994*). The issue is minimized by focusing on early segments of the ISOI response, which reports cortical organization and connectivity at columnar resolution (*Bonhoeffer and Grinvald, 1991*; *Card and Gharbawie, 2022*; *Card and Gharbawie, 2020*; *Friedman et al., 2020b*; *Lu and Roe, 2007*; *Vanzetta et al., 2004*). (2) *Task overtraining*. We started ISOI after ~2 years of task training. Extensive training as such has been shown to refine network activity and reduce metabolic demand in M1 (*Peters et al., 2014*; *Picard et al., 2013*). Both factors could have shrunk the thresholded maps from larger sizes in earlier training stages to the sizes reported here. (3) *Functional organization*. The small size of thresholded maps may reflect the functional organization of M1 and PMd. Specifically, neural activity that supports reaching and grasping could be largely concentrated in subzones of the forelimb representations. In this organizational framework, cortex in between those subzones may be better tuned for forelimb actions other than reaching and grasping. Testing this possibility would require training monkeys to perform tasks that probe a wide range of arm and hand actions, which was not done here. Nevertheless, the notion that actions may be spatially organized in cortex has support in long-train ICMS (500 ms), which maps multi-joint actions (e.g., reach, manipulate, and climb) to contiguous zones in M1 and PMd (*Baldwin et al., 2017*; *Gharbawie et al., 2011*; *Graziano et al., 2002*; *Kaas et al., 2013*).

## Functional differences between somatotopic zones

Patches that overlapped arm zones could have been tuned to different task phases than patches that overlapped hand zones. This possibility is supported by distinct time course profiles from ROIs in arm and hand zones. Nevertheless, the temporal resolution of intrinsic signals is not sufficient for relating reflectance change to behavioral events that occur in close succession (e.g., reach and grasp). We are currently conducting high-density electrophysiological recordings to interrogate relationships between movement and spiking activity in constituents of the thresholded maps. Our report on a small sample of single units recorded in other hemispheres, suggested functional differentiation between patches (*Friedman et al., 2020a*). Specifically, we found that a medial M1 patch contained more neurons tuned for reaching than neurons tuned for grasping. We found an opposite neural tuning distribution in a lateral M1 patch. Other recordings, however, from caudal M1 (central sulcus) and from rostral M1 (precentral gyrus) have reported spatially co-extensive encoding of reaching and grasping (*Rouse and Schieber, 2016*; *Saleh et al., 2012*; *Vargas-Irwin et al., 2010*). Thus, the temporal dimension provides mixed evidence about functional differences across the spatial dimension (i.e., planar axis) of the forelimb representation.

Condition differences provide insight into the functional organization of the recorded cortical activity. For example, *Nelissen and Vanduffel, 2011* defined grasp zones as regions that were more active (fMRI) in reach-to-grasp conditions as compared to a reach-only condition. A similar analysis on our data revealed patches in M1 and PMd where activity in the reach-to-grasp conditions exceeded activity in the reach-only condition (*Figure 6*). Nevertheless, these condition differences were less

apparent in the spatial size and organization of the maps, which is consistent with other studies that used univariate analyses on fMRI measurements (*Fiave et al., 2018*; *Gallivan et al., 2011*; *Nelissen et al., 2018*). In contrast, multi-variate analyses (e.g., multi-voxel pattern analysis) on the same data showed that condition identities were embedded in the spatial dimensions of activity maps.

### Differentiating between M1 and PMd activity

Functional differences between M1 and PMd were most apparent in the *observation* experiment. Time courses from the observation condition showed that there was no activity in M1. In contrast, comparable activity was recorded in PMd for the performed and observed conditions, which is consistent with the higher concentrations of mirror neurons in premotor areas than in M1 (*Grèzes et al., 2003*; *Papadourakis and Raos, 2019*; *Raos et al., 2007*). These results suggest that PMd activity could have been related to motor cognitive process as well as movement, whereas M1 activity was more closely associated with movement only.

### Conclusion

We showed that in M1 and PMd, activity related to reaching and grasping was concentrated in subzones of the forelimb representation. The results collectivity indicate that the spatial dimension of cortical motor areas is central to their functional organization. Taking this feature into consideration in electrophysiological studies can shed new light on the relationships between neural signals and movement control.

## Materials and methods

### Animals

The right hemisphere was studied in two male macaque monkeys (*Macaca mulatta*). Monkeys were 6–7 years old and weighed 9–11 kg. All procedures were approved by the University of Pittsburgh Animal Care and Use Committees (protocols 15045191 and 18042428) and followed the guidelines of the National Institutes of Health guide for the care and use of laboratory animals.

### Head post and recording chamber

After an animal acclimated to the primate chair and training environment, a head-fixation device was secured to the occipital bone and caudal parts of the parietal bone. Task training with head fixation started after ~1 month (monkey G) or ~9 months (monkey S) and lasted for ~22 months. At the end of this training period, the monkey was considered ready for optical imaging. A craniotomy was performed for implanting a chronic recording chamber (30.5 × 25.5 mm internal dimensions) over motor and somatosensory cortical areas. The chamber was secured to the skull with ceramic screws and dental cement. Within the recording chamber, the native dura was resected and replaced with a transparent silicone membrane (500 μm thickness) that we fabricated from a mold. Protocols for the artificial dura have been previously described in detail (*Arieli et al., 2002*; *Ruiz et al., 2013*). The walls of the artificial dura lined the walls of the recording chamber. The floor of the artificial dura (i.e., the optical window) was flush with the surface of cortex and facilitated visualization of cortical blood vessels and landmarks (*Figure 1A*). The walls and floor of the artificial dura delayed regrowing tissues from encroaching underneath the optical window. Single electrodes and linear electrode arrays were readily driven through the optical window without permanent deformation.

### Reach-to-grasp task

Monkeys performed a reach-to-grasp task with the left forelimb while head fixed in a primate chair. The right forelimb was secured to the waist plate. The task apparatus was positioned in front of the animal. A stepper motor rotated the carousel in between trials to present a target ~200 mm from the start position of the left hand. The relative position of the target made it directly visible and was also supposed to encourage consistent reach trajectories across conditions and trials. Task instruction was provided with LEDs mounted above the target. Photocells were embedded in multiple locations within the apparatus to monitor hand location and target manipulation. An Arduino board (Arduino Mega 2560, https://www.arduino.cc/) running a custom script (1 kHz) controlled task parameters, timing, and logged the monkey's performance on each trial. The task involved four conditions.

## Two reach-to-grasp conditions

In a successful trial, the animal had to reach, grasp, lift, and hold a sphere. The small sphere condition (12.7 mm diameter) and the large sphere condition (31.8 mm diameter) were used to motivate precision and power grips, respectively (*Figure 1D*). Spheres were attached to rods that moved in a vertical axis only. Task rules were identical for both conditions and are therefore described once. To initiate a trial, an animal placed its left hand over a photocell embedded in the waist plate. Covering the photocell for 300 ms turned on an LED, which signaled the start of the trial. Holding this start position for 5000 ms triggered the Go Cue, which was a blinking LED. The animal had 2400 ms (monkey G) or 2550 ms (monkey S) to reach, grasp, and lift the sphere; time limits were also set for each of these phases. Lifting the sphere by 15 mm turned the blinking LED solid, which signaled the end of the lift phase. Maintaining the lifted position for 1000 ms turned off the LED, which instructed the animal to release the object and return its hand to the start position within 900 ms. Maintaining the start position for 5000 ms triggered a tone and LED blinking. After an additional 2000 ms in the start position the trial was considered successful; tone and LEDs turned off and water reward was delivered. The animal could not initiate a new trial for another 3000 ms. Failure to complete any step within the allotted time window resulted in an incorrect trial signaled by a 1500-ms tone and a 5000-ms timeout in which the apparatus was unresponsive to the monkey's actions. After the timeout, a new trial could be initiated with hand placement in the start position. Across both monkeys, the median failure rate per session was 18% (IQR = 13–39%) in the precision grip condition and 16% (IQR = 9–28%) in the power grip condition.

## Reach-only condition

The target was a photocell embedded into the surface of the carousel. The photocell was visible to the monkey but was not graspable. The Go Cue was the same as the one used in the reach-to-grasp conditions, but here it prompted the monkey to reach and place its hand over the photocell. The hand had to cover the photocell for ≥220 ms (monkey G) or ≥320 ms (monkey S). All other task rules and steps were identical to the reach-to-grasp conditions. Across both monkeys the median failure rate per session was 12% (IQR = 4–32%).

## Withhold condition

In a successful trial, the monkey had to maintain its hand in the start position for ~10 s. Trial initiation was identical to the other conditions. Holding the start position for 5000 ms triggered the Withhold Cue, which was distinctly different from the Go Cue in the movement conditions. Maintaining the start position for another 2800 ms triggered a tone and LED blinking. After an additional 2000 ms in the start position the trial was considered successful and rewarded. Removing the hand from the start position at any time resulted in an incorrect trial and the same consequences described in a failed reach-to-grasp trial. Across both monkeys the median failure rate was 12% (IQR = 7–19%).

Conditions were presented in an event-related design (1 successful trial/condition/block). Condition order was randomized across blocks. We structured the trials and the inter-trial interval so that the hand would remain in the start position for ~13 s in between trials. Thus, in a successful trial from a movement condition, the arm and hand remained still in the start position for ~13 s and then moved for 1–2 s. This relatively long period without movement was useful for relating the changes in intrinsic signal (slow) to movement onset (rapid). In the withhold condition, the arm and hand were in the start position for ~10 s and no movement was allowed.

## Muscle activity

EMG was conducted in seven forelimb muscles on sessions that did not involve ISOI or other neural recordings. After head fixation, the monkey was lightly sedated with a single dose of ketamine (2–3 mg/kg, IM). Sedation was confirmed from decreased alertness and voluntary movements. At that point, pairs of stainless-steel wires (27 gauge, AM Systems) were inserted percutaneously into each muscle (~15 mm below skin). Three arm muscles were targeted (1) anterior or middle deltoid, (2) triceps brachii, and (3) biceps brachii. Four extrinsic hand muscles were targeted (1) extensor carpi radialis brevis, (2) flexor carpi radialis, (3) extensor digitorum 4–5, and (4) flexor digitorum superficialis. The task started 45–60 min after sedation. The monkey was fully alert by that point and showed

no lingering effects of sedation. The non-working forelimb was restrained and therefore could not tamper with the EMG wires.

EMG signals were filtered (bandpass 15–350 Hz) and digitized (2 kHz) using a dedicated processor (Scout model, Ripple Neuro, Salt Lake City, UT). Recorded signals were segmented into trials and their power spectral density was estimated with a discrete Fourier transform (MATLAB *fft* function, Natick, MA). Trials with power >7 $\mu V^2$ in the 1–14 Hz range were presumed to have artifact and were excluded from further analysis. EMG signals were rectified and smoothed with a 100-ms sliding window (MATLAB *filtfilt* function). Finally, for each muscle in each movement condition, the average signal was computed across trials and sessions (308–595 trials/muscle).

## Joint kinematics

Forelimb joints were tracked in 3D on sessions that did not involve ISOI or other neural recordings. Before the task started, six LEDs were secured to sites on the left forelimb to track up to two joints/ session. A motion capture system (Impulse X2, Phasespace) outfitted with six cameras recorded LED positions and logged *x*, *y*, *z* coordinates (480 Hz). For each tracked joint, LEDs were configured to form imaginary vectors or planes. For example, to track the elbow, three LEDs were placed in a triangular formation on the upper arm (i.e., parallel to the humerus) and another three-LED formation was placed on the forearm (i.e., parallel to the ulna). Elbow flexion/extension was calculated as the angle between the vector aligned with the upper arm and the vector aligned with the forearm. Pronation/ supination was calculated as the angle between the normal of the plane of the upper arm and the normal of the plane of the forearm. A similar approach was adopted for (1) shoulder flexion/extension, (2) shoulder abduction/adduction, (3) wrist flexion/extension, and (4) digits flexion/extension. Time series of LED coordinates were segmented into trials. Trials were excluded from analyses if LED positions were not logged for >2% of trial duration. Such dropouts were typically due to LED occlusion by the forelimb, primate chair, or grasp apparatus. Kinematic profiles were calculated trial-by-trial and then averaged for each condition (143–405 trials/degree of freedom).

## Movement quantification

EMG and kinematics were quantified trial-by-trial. Time-resolved traces were generated for each recorded muscle and degree of freedom. The AUC was calculated with trapezoidal approximation (MATLAB *trapz* function) applied to the time-resolved traces. We focused on the period from Cue onset until the end of forelimb withdrawal when the hand returned to the start position. For each muscle and degree of freedom, AUC values were standardized across task condition. Standardization was done separately for each session to account for recording variations across sessions. The average AUC for a muscle, or a joint, was calculated as the median of all trials acquired for a task condition. The average AUC for a group of muscles, or a group of joints, was calculated as a mean weighted by the number of trials.

## Motor mapping

We used ICMS to map the somatotopic organization of frontal motor areas. In monkey S, all sites (*n* = 158) were investigated with a microelectrode in dedicated motor mapping sessions. We used the same approach in >50% of the sites (*n* = 118) in monkey G. The remaining sites (*n* = 99) were mapped with a linear electrode array at the end of electrophysiological recordings that will be presented in a separate report. In the dedicated motor mapping sessions, the monkey was head fixed in the primate chair and sedated (ketamine, 2–3 mg/kg, IM, every 60–90 min). This mild sedation reduced voluntary movements but did not suppress reflexes or muscle tone. A hydraulic microdrive (Narishige MO-10) connected to a customized three-axis micromanipulator was attached to the recording chamber for positioning a tungsten microelectrode [250 $\mu$m shaft diameter, impedance = 850 ± 97 k$\Omega$ (mean ± standard deviation, SD)] or a platinum/iridium microelectrode [250 $\mu$m shaft diameter, impedance = 660 ± 153 k$\Omega$ (mean + SD)]. A surgical microscope aided with microelectrode placement in relation to cortical microvessels. The microelectrode was in recording mode at the start of every penetration. Voltage differential was amplified (10,000×) and filtered (bandpass 300–5000 Hz) using an AC Amplifier (Model 2800, AM Systems, Sequim, WA). The signal was passed through a 50/60 Hz noise eliminator (HumBug, Quest Scientific Instruments Inc) and monitored with an oscilloscope and a loudspeaker. As the electrode was lowered, the first evidence of neural activity was considered 500 $\mu$m below the

pial surface. The microelectrode was then switched to stimulation mode and the effects of ICMS were tested at $\geq$4 depths (500, 1000, 1500, and 2000 μm). Microstimulation trains (18 monophasic, cathodal pulses, 0.2 ms pulse width, 300 Hz) were delivered from an 8-Channel Stimulator (model 3800, AM Systems). Current amplitude, controlled with a stimulus isolation unit (model BSI-2A, BAK Electronics), was increased until a movement was evoked (max 300 μA). The stimulation threshold for each depth was the current amplitude that evoked movement on 50% of stimulation trains.

One experimenter controlled the location and depth of the microelectrode. A second experimenter, blind to microelectrode location, controlled the microstimulation. Both experimenters inspected the evoked response and discussed their observations to reach consensus about the active joints (i.e., digits, elbow, etc.) and movement type (flexion, extension, etc.). Movement classification was not cross-checked against EMG recording or motion tracking. The overall classification for a given penetration included all movements evoked within 30% of the lowest threshold across depths. The location of each penetration (500–1000 μm apart) was recorded in relation to cortical microvessels. Color-coded maps were generated from these data using a voronoi diagram (MATLAB *voronoi* function) with a maximum tile radius of 750 μm (*Figure 2B, F*). The rostral border of M1 was marked to separate sites with thresholds <30 μA from higher threshold sites (*Figure 2C, G*).

The mapping approach was similar for penetration sites stimulated with a linear electrode array (32 or 24 channels, 15 μm contact diameter, 100 μm inter-contact distance, 210–260 μm probe diameter; V-Probe, Plexon). Each penetration was mapped ~2.5 hr after the linear array was inserted into cortex, which was also the end of electrophysiological recordings during task performance. Only 1 penetration was mapped per session. Microstimulation parameters were identical to the ones used with the microelectrode but were controlled here using the Trellis Software Suite (Ripple Neuro). Channels were stimulated one at a time and every other channel was used. One experimenter controlled the microstimulation and classified the evoked movements.

## Intrinsic signal optical imaging

The FOV was illuminated with three independently controlled red LEDs (630 nm wavelength). Each LED was outfitted with a lens to diffuse the light emitted. The experimenters optimized LED positions and brightness with the aid of a real-time heatmap of the FOV. Camera frames 768 × 768 pixels (monkey G) or 1080 × 1310 pixels (monkey S) were captured with a 12-bit CMOS sensor (Photon Focus, Lachen, Switzerland). The tandem lens combination achieved a FOV of 15 × 15 or 22 × 26 mm, both at 20 μm²/pixel. Frames were temporally averaged from 100 to 10 frames/s then saved. Image acquisition and parameters were controlled with an Imager 3001 system (Optical Imaging Ltd, Rehovot, Israel). In every trial, imaging started 1 s before Cue onset and lasted for 7 s unless otherwise stated. For spatial reference, a high contrast image of the cortical surface was captured with green illumination (528 nm wavelength) at the start of every session.

## Image processing

Image processing was conducted on individual imaging sessions. Data frames were rigid aligned in *x*, *y* coordinates (MATLAB *estimateGeometricTransform* function) to a reference frame from the middle of the session. A trial was excluded if any frame was out of register by >10 pixels (i.e., >200 μm). In the remaining trials, the first 10 frames (−1.0 to 0 s from Cue) of a trial were averaged then subtracted from all frames in the same trial. This subtraction converted pixel values to reflectance change with respect to baseline, which effectively normalized every trial to itself. To correct uneven illumination and residual motion artifact, frames were processed with a high-pass median filter (kernel = 250 pixels). A low-pass Gaussian filter (kernel = 5 pixels) was used for smoothing. To accelerate the spatial filtering computations, frames were temporarily down sampled by a factor of 4.

Frames from different sessions were aligned to a common reference, which was a high contrast image of the cortical surface. For every imaging session, we marked 25–60 points that were apparent in the microvessel patterns from that session and in the common reference. These points were used to construct a mesh grid for the reference and session images. Non-rigid transformation (multilevel B-spline approximation) was then applied to fit the session mesh grid to the reference mesh grid (*Koon, 2008*; *Lee et al., 1997*; *Rueckert et al., 1999*). The transformation matrix was then used to co-register all frames from the session to the common reference. Frames could then be averaged within a session and across sessions. To enhance visualization of average frames, the distribution of

pixel values was clipped in relation to the median pixel value. Clipping was excluded, however, from time courses and thresholded maps.

## Reflectance change time course

Time courses were generated from two types of ROIs. (1) Small circles (radius = 20 pixels [0.4 mm]) that were placed in arm zones and hand zones in M1 and PMd. (2) A larger ROI that included the M1 and PMd forelimb representations. In both cases, pixel values within an ROI were averaged to obtain 1 value per frame. Pixels that overlapped blood vessels were excluded. Time courses were based on trial-averaged time series.

In the present illumination (630 nm, i.e., red), pixel darkening is accepted as a lagging indicator of neural activity (*Grinvald et al., 1986*). The increased consumption of oxygen by local neural activity is believed to increase deoxyhemoglobin concentrations, which absorbs red light and is therefore detected as pixel darkening (*Malonek and Grinvald, 1996*; *Shtoyerman et al., 2000*). Some have likened the early pixel darkening in ISOI to the initial dip in BOLD fMRI (*Ances, 2004*; *Kim et al., 2000*; *Menon et al., 1995*). Nevertheless, there is evidence from multi-spectral imaging to suggest that pixel darkening in ISOI is a more complicated response driven by changes in total hemoglobin, blood volume, and blood flow (*Sirotin et al., 2009*). In contrast, pixel brightening reports increase in the concentration of oxygenated hemoglobin and blood flow/volume and may therefore resemble the increase in BOLD fMRI (*Chen-Bee et al., 2007*).

## Thresholded activity maps

To identify pixels that darkened in response to task-related movements, we compared frames from the end of movement (i.e., movement frames) to frames acquired before movement onset (i.e., baseline frames). Every trial contributed 1 movement frame and 1 baseline frame (8 sessions, 236–432 trials/condition). A baseline frame was an average of the 10 data frames acquired from −1.0 to Cue. Depending on the analysis, a movement frame was an average of 39 data frames, or an average of 5 data frames, acquired after movement completion.

A two-sample *t*-test was then conducted pixel-by-pixel to compare values between movement frames and baseline frames. Right tail values (p < 1e−4) indicated that the affiliate pixels brightened significantly in movement frames as compared to baseline frames. Left tail values (p < 1e−4) indicated that pixels darkened significantly in movement frames as compared to baseline frames. Pixels were also thresholded after Bonferroni correction for multiple comparisons (p < 1e−7); $\alpha$ here was based on the number of pixels (excluding blood vessels) in the M1 and PMd forelimb representations. For both threshold levels, pixels that darkened significantly were considered the constituents of the thresholded activity map. To denoise those maps, significant pixels were removed if they did not belong in a group with ≥10 connected significant pixels.

## Thresholded map quantification

Thresholded activity maps and the motor maps were both registered to the common reference. Pixels within the M1 and PMd forelimb representations were counted. Those pixel counts were then expressed as a percentage of the total number of pixels that make up the forelimb representations. The analysis excluded pixels that overlapped major blood vessels.

## Paired comparisons of condition maps

Cortical activity was directly compared between condition pairs in two ways. (1) Two sample *t*-test as described earlier but deployed here for comparing movement frames between conditions. (2) Cross-correlation between condition pairs. For each condition, an average time series (*n* = 70 frames) was generated from all trials. Each average frame was then converted into a one-dimensional array. Pairs of arrays from different conditions but matched time points, were cross-correlated at zero lag (e.g., *power grip condition Frame 1 × reach-only condition Frame 1*). Thus, the full comparison between two conditions returned 70 correlation coefficients, but we only focused on the coefficients from the post-movement frames (n = 39). We used the motor map to constrain the analysis to the M1 and PMd forelimb representations.

## Statistical analyses

All analyses were done in MATLAB. Function names are in parentheses. We used parametric tests after verifying that our data met parametric assumptions. Specifically, we confirmed that the variance

was comparable between groups (*vartestn*) and that the data were normally distributed, which we established from skewness (*skewness*) and kurtosis (*kurtosis*) values between −1 and +1. We used paired *t*-tests (*ttest2*) for the pixel-by-pixel comparisons used to generate thresholded activity maps. We used one-way ANOVA (*anova1*) for condition comparisons of imaging time series and metrics of forelimb use. Post hoc comparisons (*multcompare*) were follow-up *t*-tests corrected by the number of comparisons.

## Acknowledgements

Project was supported with funds from NIH (R01 NS105697), Whitehall Foundation (2017-12-94), and University of Pittsburgh Brain Institute. We are grateful to ToniAnn Zullo for outstanding animal care before, after, and during, all procedures. We thank Dr. Nick Card for advice on analyses.

## Additional information

### Funding

| Funder | Grant reference number | Author |
| --- | --- | --- |
| National Institutes of Health | R01 NS105697 | Omar A Gharbawie |
| Whitehall Foundation | 2017-12-94 | Omar A Gharbawie |

The funders had no role in study design, data collection, and interpretation, or the decision to submit the work for publication.

### Author contributions

Nicholas G Chehade, Resources, Data curation, Software, Formal analysis, Validation, Investigation, Visualization, Methodology, Writing – original draft, Writing – review and editing; Omar A Gharbawie, Conceptualization, Resources, Data curation, Formal analysis, Supervision, Funding acquisition, Validation, Investigation, Visualization, Methodology, Writing – original draft, Project administration, Writing – review and editing

### Author ORCIDs

Nicholas G Chehade http://orcid.org/0000-0002-1296-948X
Omar A Gharbawie https://orcid.org/0000-0002-2744-9305

### Ethics

All procedures were approved by the University of Pittsburgh Animal Care and Use Committees (protocol #21049001) and followed the guidelines of the National Institutes of Health guide for the care and use of laboratory animals.

### Decision letter and Author response

Decision letter https://doi.org/10.7554/eLife.83196.sa1
Author response https://doi.org/10.7554/eLife.83196.sa2

## Additional files

### Supplementary files

• MDAR checklist

### Data availability

All data and code used in this paper are posted on OSF (https://osf.io/7sgbe/).

The following dataset was generated:

| Author(s) | Year | Dataset title | Dataset URL | Database and Identifier |
|---|---|---|---|---|
| Chehade N | 2023 | Motor actions are spatially organized in motor and dorsal premotor cortex | https://doi.org/10.17605/OSF.IO/7SGBE | Open Science Framework, 10.17605/OSF.IO/7SGBE |

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
