## [Editor Report]

This valuable and technically highly demanding paper combines intra-cortical stimulation and large-field-of view optical imaging to study the forelimb representation in two macaque monkeys. The authors provide convincing evidence that reach-to-grasp and reach-only tasks only activated restricted subset of the forelimb area (as revealed through stimulation). While these results are consistent with the idea of clusters of neural activity that correspond to different forelimb actions, the evidence that this particular claim, as the discussion points out, remains incomplete.

---

## [Decision Letter]

**Decision letter after peer review:**

Thank you for submitting your article "Neural activity is spatially clustered in motor and dorsal premotor cortex" for consideration by *eLife*. Your article has been reviewed by 2 peer reviewers, one of whom is a member of our Board of Reviewing Editors, and the evaluation has been overseen by Joshua Gold as the Senior Editor. The following individual involved in the review of your submission has agreed to reveal their identity: Wim Vanduffel (Reviewer #2).

Essential revisions:

1) The evidence for "clustered" activity is incomplete. It is not clear whether there is convincing evidence that only a small part of the arm/hand representation is activated. Showing that the non-active regions can be activated by other actions would have been the most direct and convincing evidence here (see main comment 1. from both reviewers 1 and 2).

2) Both reviewers also point out that the comparison between activation and stimulation data (which is the unique feature of this dataset) is rather superficial and both reviewers saw a lot of potential here.

*Reviewer #1 (Recommendations for the authors):*

This paper describes the neural activity, measured by intrinsic optical imaging in reach-to-grasp, and reach-only conditions in relation to the Intra-cortical micro stimulation maps. The paper mostly describes a relatively unique and potentially useful data set. However, in the current version, no real hypotheses about the organization of M1 and PMd are tested convincingly. For example, the claim of "clustered neural activity" is not tested against any quantifiable alternative hypothesis of non-clustered activity, and support for this idea is therefore incomplete.

The combination of intrinsic optical imaging and intra-cortical micro-stimulation of the motor system of two macaque monkeys promised to be a unique and highly interesting dataset. The experiments are carefully conducted. In the analysis and interpretation of the results, however, the paper was disappointing to me. The two main weaknesses in my mind were:

The alternative hypotheses depicted in Figure 1B are not subjected to any quantifiable test. When is an activity considered to be clustered and when is it distributed? The fact that the observed actions only activate a small portion of the forelimb area (Figure 5G, H) is utterly unconvincing, as this analysis is highly threshold-dependent. Furthermore, it could be the case that the non-activated regions simply do not give a good intrinsic signal, as they are close to microvasculature (something that you actually seem to argue in Figure 6b). Until the authors can show that the other parts of the forelimb area are clearly activated for other forelimb actions (as you suggest on line 625), I believe the claim of cluster neural activity stands unsupported.The most interesting part of the study (which cannot be easily replicated with human fMRI studies) is the correspondence between the evoked activity and intra-cortical stimulation maps. However, this is impeded by the subjective and low-dimensional description of the evoked movement during stimulation (mainly classifying the moving body part), and the relatively low-dimensional nature (4 conditions) of the evoked activity.Many details about the statistical analysis remain unclear and seem not well motivated.

Figure 1B: What does FL (gray area, caption) stand for? I can't see that the term is defined.

Line 158: I gather that unsuccessful trials were repeated. What were the repetition rates in the three conditions? Was the big and small sphere (here listed as one condition) also varied in a pseudorandom manner?

Line 223: I gather from this that the movement elicited in the ICMS was judged qualitatively by the two experimenters, but no EMG / movement kinematics was recorded as in the overt behavior. Did you make any attempts to validate the experimenter's judgement against objective recordings? Could you calculate inter-rater reliability or did the two experimenters interact with each other during the judgement?

Line 260: The baseline was either the first frame or the initial first four frames. How did you decide when to use which method? The statement "to increase signal-to-noise ratio" is somewhat cryptic. Increase in respect to what? Baseline subtraction clearly needs to be done in some form or the other. If you wanted a stable baseline, why not take the entire period before cue appearance?

Line 264-266. When you state: kernel = 250 or 550 pixels and low pass filer 5 or 15 pixels does this refer to the data from the two different monkeys? How were these values decided?

Line 278: If you do another baseline subtraction here, is the baseline subtraction on line 260 not superfluous? When did you use 5 and when 9 frames?

For the fMRI crowd, could you add here how pixel darkening and brightening relate to blood flow changes and blood oxygenation changes?

Line 305: Your nomenclature of cells / ROIs is somewhat confusing. What was the motivation to spatially average pixels across each cell/ROI instead of entering all pixels into the clustering analysis? What do mean by "Spatially matched cells"? Was co-registering not something that was done on the entire window?

Line 312: How big was the grid in horizontal and vertical dimensions for each monkey? Is this where the 1700 / 2249 numbers come from?

Line 313: A distances metric is defined to be zeros to itself, dist(A,A) = 0. The correlation of A with itself is 1. Do you mean to say that you used 1-max_correlation, or that you used the correlation as a similarity metric?

Line 323: My guess the point of this analysis is that the total distances decrease with an increasing number of clusters (and the correlation increases with an increasing number of clusters). So, fitting a line assumes a linear relationship between these two variables. When you say "The longest of those orthogonal lines identified the optimal number of clusters" why are the lines orthogonal? By longest – do you mean the largest deviation between the linear fit either above or below the line? This would pick either an especially good or bad parcellation. Or did you look at deviations above or below only?

Line 327: In the methods, we have 3 conditions – in the results 4. Which one is correct?

Line 370: Is the darkening appearing at 1s an artifact of the measurement or real? At least for the posterior border, it looks like it follows exactly the border of the window.

Line 451: "that shifted and expanded across the time series" – is there a statistical test for the claim that the activity region shifted?

Line 462: To what degree is the fact the activity averaged over the entire window remains stable simply a consequence of the high-pass filtering applied to each image? High-pass filtering basically removes the mean across the entire window, no?

Line 466: What is meant by "domains"? Activity clusters?

Line 5E: The analysis here involves statistical testing within each session, and then averaging activity estimates across sessions, subject to an arbitrary 50% threshold for statistical significance within the session. I cannot see a good motivation for this awkward type of analysis. If you want to consider trials a random effect, I would recommend calculating a statistical test for all trials across sessions combined. If you want to consider sessions as a random effect, then calculate a t-test across the 8 repeated activity estimates per session – this analysis also automatically takes the variability across trials into account. Or is there another reason why you chose this baroque style of analysis that is mentioned?

Line 474: "organization of domains was more similar across conditions within an animal than for the same condition across animals". See Ejaz et al. (2015). Nature Neuroscience, for a similar observation in human fMRI motor maps.

Line 483: The overlap analysis is unfortunately quite dependent on the threshold.

Line 494: form → from?

I am not sure what I am supposed to learn from the cluster analysis. It is entitled "Clustering time courses recapitulates spatial patterns of activity" – likely referring to the similarity between Figure 6a and 6b. As the clustering partly depends on the magnitude of the signal at the time points presented in Figure 6a, what is the alternative? Is this similarity more than expected by chance?

Line 527-546. The overlap between the maps is very hard to interpret, as it is highly dependent on the threshold procedure applied, which is somewhat arbitrarily chosen (see above). Even random maps of a certain size would overlap to some degree, and even identical "true" maps don't overlap completely due to measurement error. An overlap of 0.5 has no intrinsic meaning. It would probably be more informative to report the correlation between unthresholded maps, and compare this to a noise ceiling, where you correlate the maps of one condition across sessions.

*Reviewer #2 (Recommendations for the authors):*

Chehade and Gharbawie investigated motor and premotor cortex in macaque monkeys performing grasping and reaching tasks. They used intrinsic signal optical imaging (ISOI) covering an exceedingly large field-of-view extending from the IPS to the PS. They compared reaching and fine/power-grip grasping ISOI maps with "motor" maps which they obtained using extensive intracranial microstimulation. The grasping/reaching-induced activity activated relatively isolated portions of M1 and PMd, and did not cover the entire ICM-induced 'motor' maps of the upper limbs. The authors suggest that small subzones exist in M1 and PMd that are preferentially activated by different types of forelimb actions. In general, the authors address an important topic. The results are not only highly relevant for increasing our basic understanding of the functional architecture of the motor-premotor cortex and how it represents different types of forelimb actions, but also for the development of brain-machine interfaces. These are challenging experiments to perform and add to the existing yet complementary electrophysiology, fMRI, and optical imaging experiments that have been performed on this topic – due to the high sensitivity and large coverage of the particular IOSI methods employed by the authors. The manuscript is generally well written and the analyses seem overall adequate – but see below for some additional analyses that should be done. Although I'm generally enthusiastic about this manuscript, there are two major issues that should be clarified. These major questions relate mainly to potential thresholding issues and clustering issues.

1) The main claim of the authors is that specific forelimb actions activate only a small fraction of what they call the motor map (i.e., those parts of M1/PMd that evoke muscle contractions upon ICM). The action-related activity is measured by ISOI. When looking a the 'raw' reflectance maps, it is rather clear that relatively wide portions of the exposed cortex are activated by grasping/reaching, especially at later time points after the action. In fact, another reading of the results may be that there are two zones of 'deactivation' that split a large swath of motor-premotor cortex being activated by the grasping/reaching actions. (e.g. at 6 seconds after the cue in Figure 3A, 5A). At first sight, the 'deactivated' regions seem to be located in the cortex representing the trunk/shoulder/face – hence regions not necessarily activated (or only weakly) during the grasping/reaching actions. If true, this means that most of the relevant M1/PMd cortex IS activated during the latter actions – opposing the 'clustering' claims of the authors. This raises the question of whether the 'granularity' claimed by the authors is:

Threshold dependent. In this context, the authors should provide an analysis whereby 'granularity' is shown independent of statistical thresholds of the ISOI maps.Dependent on the time-point one assesses the maps. Given the sluggish hemodynamic responses, it is unclear which part of the ISOI maps conveys the most information relative to the cue and arm/hand movements. I suspect that timepoints > 6 s will reveal even larger 'homogeneous' activations compared to the maps < 6s.

In fact, Figure 5F (which is highly thresholded) shows a surprisingly good match between the different forelimb actions, which argues against the existence of small subzones that are preferentially activated by different types of forelimb actions -the main claim of the authors.

2) Related to the previous point, the ROI selections/definitions for the time course analyses seem highly arbitrary. As indicated in the introduction, the clustering hypothesis dictates that "an arm function would be concentrated in subzones of the motor arm zones. Neural activity in adjacent subzones would be tuned for other arm functions." To test this hypothesis directly in a straightforward manner, the authors could use the results from the ICM experiment to construct independent ROIs and to evaluate the ISOI responses for the different actions. In that case, the authors could do a straightforward ANOVA (if the data permits parametric analyses) with ROI, action, and time point (and possibly subject) as factors.

3) More details about the transparent silicone membrane should be provided: How implanted? How maintained? Please provide pictures at the end of the 9 and 22months periods.

4) Figure 1D: Are the yellow circles task cues that the monkeys saw? If not, what were the different cues (Reach, Grasp, Withhold)? Were precision and power grip (and reach?) trials similarly cued?

5) For the MEG activity a total of 14 27gauge wires were inserted in 7 muscles. This sounds rather invasive. Did monkeys tolerate this easily? Were local anesthetics used? How deep were the wires inserted in the muscle? How did one prevent the monkeys grabbing these wires?

6) The large FOV was illuminated using 2-3 LEDs. Was illumination uniform throughout the FOV? If not, can this lead to inhomogeneous sensitivity?

7) Pixel values were clipped, if I understood correctly, at either 0.3 SD or 1 SD from the mean. This significantly changes the dynamic range of these pixel values. Are results different without clipping?

8) An arbitrarily p<0.0001 statistical threshold was used. Why was no correction for multiple correspondence performed as one collects data from ~ 600k-1400k pixels (yet without considering the masks)?

9) What is the reason for obvious edge effects in the withhold condition (Figure 3A, 3F) and apparently also the precision grip condition (Figure 5A: 1.1-3.5sec)? Is this artifactual?

10) How are the ROIs defined (cfr major point 2)? They differ in size and location between analyses (e.g. Figure 3 vs Figure 5/S1). Please discuss.

11) I find it a bit counter-intuitive that the same color-code (i.e. black values) is given for 'activations' in panel 3G and 'deactivations' in panel 3H. Why not using red and blue instead? Along the same vein: activations in the time-courses are negative (reflecting the darkening). It would be instructive for the non-experienced reader to either add activation/ suppression on the figures (above and below zero in Figures 3I and J and 5C), or to invert the Y-axis.

12) Line 426: Figure 5K does not exist.

13) Line 520: It is argued that the unsupervised cluster analysis, which is quite interesting, is similar in both monkeys. However, this is not obvious from the data: neither in the spatial domain (Figure 6B vs S2B) nor in the temporal domain Figure 6C and S2 C, especially the blue plots. In fact, the clustering data from monkey G reveal a rather widespread, uninterrupted pattern arguing against the 'cluster hypothesis' of the authors. This should be discussed in more depth.

14) Figure 7A: It is unclear which analysis was done. Is this simply giving 3 different colors to each pixel – indicating 1) precision > baseline; 2) reach > baseline 3) both > baseline (for the left panel)? Why not performing straight (pair-wise) subtractions?

---

## [Author Response]

Essential revisions:1) The evidence for "clustered" activity is incomplete. It is not clear whether there is convincing evidence that only a small part of the arm/hand representation is activated. Showing that the non-active regions can be activated by other actions would have been the most direct and convincing evidence here (see main comment 1. from both reviewers 1 and 2).

We appreciate the reviewers’ concerns. We overhauled that analyses for measuring the spatial extent of cortical activity. We present new and more robust evidence that cortical activity overlapped relatively small portions (<40%) of M1 and PMd forelimb representations (Figure 5 and Figure 5 – supplemental Figure 1-2).

We agree with the reviewers; had the non-active regions shown activation in other arm/hand actions, that would have supported the notion of functionally specific subregions. Obtaining that data would require optical imaging while monkeys perform a variety of tasks that test a range of arm and hand movements (e.g., wide range of finger movements, wrist postures, arm directions, etc.). The two monkeys that supplied data for this study were trained only on a reach-to-grasp task with few task conditions. Moreover, the monkeys are no longer available for optical imaging. Thus, obtaining the desired data would involve conducting a larger version of the present study in naïve monkeys. A conservative estimate for completing such a study is 2-3 years. Until we can complete such a study, we expanded the Discussion to consider limitations of the behavioral task and our interpretation of the results.

2) Both reviewers also point out that the comparison between activation and stimulation data (which is the unique feature of this dataset) is rather superficial and both reviewers saw a lot of potential here.

We agree with the reviewers’ concerns. We now leverage each monkey’s motor map to guide placement of regions-of-interest (ROIs) for measuring time courses. Similarly, we use the motor map to spatially constrain the field-of-view for analyses of the activity maps.

Reviewer #1 (Recommendations for the authors):This paper describes the neural activity, measured by intrinsic optical imaging in reach-to-grasp, and reach-only conditions in relation to the Intra-cortical micro stimulation maps. The paper mostly describes a relatively unique and potentially useful data set. However, in the current version, no real hypotheses about the organization of M1 and PMd are tested convincingly. For example, the claim of "clustered neural activity" is not tested against any quantifiable alternative hypothesis of non-clustered activity, and support for this idea is therefore incomplete.The combination of intrinsic optical imaging and intra-cortical micro-stimulation of the motor system of two macaque monkeys promised to be a unique and highly interesting dataset. The experiments are carefully conducted. In the analysis and interpretation of the results, however, the paper was disappointing to me. The two main weaknesses in my mind were:a) The alternative hypotheses depicted in Figure 1B are not subjected to any quantifiable test. When is an activity considered to be clustered and when is it distributed? The fact that the observed actions only activate a small portion of the forelimb area (Figure 5G, H) is utterly unconvincing, as this analysis is highly threshold-dependent. Furthermore, it could be the case that the non-activated regions simply do not give a good intrinsic signal, as they are close to microvasculature (something that you actually seem to argue in Figure 6b). Until the authors can show that the other parts of the forelimb area are clearly activated for other forelimb actions (as you suggest on line 625), I believe the claim of cluster neural activity stands unsupported.

We appreciate the reviewer’s concerns and we have made several revisions.

(1) The two panels in Figure 1B should have been presented as potential outcomes as opposed to hypotheses in need of quantifiable testing. We revised the Introduction (line 105-111) and the Results (line 149-152) accordingly.

(2) We agree that the thresholding procedure adopted in the original submission could have impacted the spatial measurements of cortical activity (i.e., Figure 5G-H in original submission). We have completely revised the thresholding procedure and it is now based on statistical comparisons that include all trials (instead of thresholding by number of sessions in the original submission). Thus, the thresholded maps in Figure 5G & 5J are now obtained from pixel-by-pixel comparisons (t-tests, p<1e-4) between frames acquired post-movement and frames acquired before movement. Nevertheless, even with this relatively relaxed threshold, the largest activity maps overlapped <40% of the forelimb representations.

It is important to note that major vessels were excluded from the thresholded map and from the motor map. Thus, uncertainty about imaging in and around vessels was likely not a factor in the calculated overlap between thresholded maps and the motor map.

(3) We agree that showing activation in other parts of the forelimb representations in response to action other than reach-to-grasp would have supported some of the arguments that we previously put forth. Unfortunately, we do not have the supporting data and obtaining it would take months/years. We have therefore expanded the Discussion to include limitations of the behavioral task (line 439-443).

b) The most interesting part of the study (which cannot be easily replicated with human fMRI studies) is the correspondence between the evoked activity and intra-cortical stimulation maps. However, this is impeded by the subjective and low-dimensional description of the evoked movement during stimulation (mainly classifying the moving body part), and the relatively low-dimensional nature (4 conditions) of the evoked activity.

We agree with the reviewer on all accounts. We expanded the Discussion to consider the low dimensionality of the motor maps and the behavioral task (line 439-449).

Measuring cortical activity in a variety of motor tasks would likely have provided additional insight about movement-related cortical activity. Nevertheless, including additional tasks, even if it were possible to do so in the same monkeys, would have delayed study completion by months/years. The hidden challenge of the experimental design is that each monkey is trained to not move for many seconds to minimize contamination of ISOI signals. For example, from trial initiation to Go Cue, the monkey must hold its hand in the start position for 5 seconds. Similarly, after movement completion, the monkey must hold its hand in the start position for another 5 seconds. In between successful trials, a monkey must wait for ~12 seconds before it can initiate a new trial. These durations are >1 order of magnitude longer than in electrophysiological studies in comparable tasks. Achieving consistent task performance with the long durations used here, took months of daily training. Moreover, our monkeys typically run out of steam after ~60-70 min of working on the task. This forces us to limit the overall number of task conditions tested in a session, to obtain a large enough number of trials from each condition.

c) Many details about the statistical analysis remain unclear and seem not well motivated.

We address the reviewer’s specific concerns below.

Figure 1B: What does FL (gray area, caption) stand for? I can't see that the term is defined.

FL is short for forelimb. The term is now defined in the figure caption.

Line 158: I gather that unsuccessful trials were repeated. What were the repetition rates in the three conditions? Was the big and small sphere (here listed as one condition) also varied in a pseudorandom manner?

All conditions, including the large and small spheres, were presented in pseudorandom order (line 624-625). We revised the Methods to clarify that the large and small sphere are considered separate conditions (line 593-594).

We calculated the average number of repetitions for each condition. Results are pooled across monkeys.

Median (IQR)Reach-to-grasp with precision grip: 0.6 (0.2 – 1.3)Reach-to-grasp with power grip: 0.4 (0.2 – 0.7)Reach only: 0.3 (0.1 – 0.7)Withhold: 0.3 (0.1 – 0.4)

We also calculated the average failure rate for each condition and report those numbers in the Methods (line 607-609; 615; 622-623). Results are pooled across monkeys.

Median (IQR)Reach-to-grasp with precision grip: 18%; (13 – 39%)Reach-to-grasp with power grip: 16%; (9 – 28%)Reach only: 12%; (4 – 32%)Withhold: 12%; (7 – 19%)

Line 223: I gather from this that the movement elicited in the ICMS was judged qualitatively by the two experimenters, but no EMG / movement kinematics was recorded as in the overt behavior. Did you make any attempts to validate the experimenter's judgement against objective recordings? Could you calculate inter-rater reliability or did the two experimenters interact with each other during the judgement?

In previous experiments in macaques and squirrel monkeys, we benchmarked experimenter classification of ICMS-evoked movement against ICMS-evoked EMG. In winner-take-all classification like the one adopted here; experimenter classification of evoked movements were consistent with the largest EMG response. Nevertheless, that benchmarking was not redone for this study. Also, inter-rate reliability cannot be assessed because the experimenters collaborated on movement classification. We expanded the Methods (line 708-711) and Discussion to reflect these limitations (line 443-449).

Line 260: The baseline was either the first frame or the initial first four frames. How did you decide when to use which method? The statement "to increase signal-to-noise ratio" is somewhat cryptic. Increase in respect to what? Baseline subtraction clearly needs to be done in some form or the other. If you wanted a stable baseline, why not take the entire period before cue appearance?

We agree with the reviewer on all points. First-frame subtraction has now been standardized across monkeys and imaging sessions. The first frame is now an average of 10 frames acquired from -1.0 to 0 s from Cue. We revised all analyses and figures accordingly and noted little/no effect on the results. Also, for clarity, we revised the statement about signal-to-noise ratio (line 745-748).

Line 264-266. When you state: kernel = 250 or 550 pixels and low pass filer 5 or 15 pixels does this refer to the data from the two different monkeys? How were these values decided?

Yes, in the original submission, kernel sizes differed between monkeys. We have reprocessed the data so that filtering parameters are identical for both monkeys. High pass kernel = 250 pixels. Low pass kernel = 5 pixels (line 748-750).

Line 278: If you do another baseline subtraction here, is the baseline subtraction on line 260 not superfluous? When did you use 5 and when 9 frames?

We revised the first-frame subtraction (i.e., baseline subtraction) to ensure that it is done consistently across sessions and monkeys (line 745-748).

We agree that a second baseline subtraction would make the first one superfluous. But only one subtraction was done: the first-frame was subtracted from all subsequent frames in the same trial (line 260 in original submission). After first-frame subtraction, we average baseline frames and separately average movement frames, to statistically compare them to each other.

For the fMRI crowd, could you add here how pixel darkening and brightening relate to blood flow changes and blood oxygenation changes?

We expanded this section to describe pixel darkening and brightening and to relate it to hemodynamic signals recorded in fMRI (line 769-778). In brief, for the present illumination (630 nm wavelength), pixel darkening is established as an indicator of increased deoxyhemoglobin (HbR) concentrations. Pixel darkening in ISOI has been likened to the “initial dip” in BOLD fMRI, which has been elusive in some fMRI studies. In contrast pixel brightening in ISOI is considered an indicator of increased concentrations of oxygenated hemoglobin (HbO) and blood volume/flow. ISOI pixel brightening has therefore been likened to an increase in BOLD fMRI.

Line 305: Your nomenclature of cells / ROIs is somewhat confusing. What was the motivation to spatially average pixels across each cell/ROI instead of entering all pixels into the clustering analysis? What do mean by "Spatially matched cells"? Was co-registering not something that was done on the entire window?

W averaged pixels within a cell for: (1) denoising, and (2) speeding up computation. Yes, co-registration was done for the entire field-of-view. We should have referred to that concept instead of the confusing “spatially matched cells”.

We agree with the reviewer that aspects of the clustering analysis, including nomenclature, were confusing. We redid the clustering analysis and believe that the revisions would have addressed the points raised here and elsewhere by both reviewers. Nevertheless, we decided to replace the clustering analysis with a more straightforward analysis that is presented (line 345-374 and Figure 6).

The purpose of clustering was to extract spatial patterns of activity from time series. We were enthusiastic about clustering because of its potential to return meaningful activity maps without input from us about frame range. To partially preserve that feature in the new analysis, we used all frames acquired post movement (i.e., 39 frames captured from +2.2 to +6.0s from Cue). We chose this frame range to (1) exclude frames that may contain motion artifact, and (2) focus on frames likely to have task-related activity.

Line 312: How big was the grid in horizontal and vertical dimensions for each monkey? Is this where the 1700 / 2249 numbers come from?

This inquiry pertains to the clustering analysis, which has been removed from the manuscript.

Nevertheless, yes, the numbers refer to the dimensions of the grid. The grids in both monkeys had identical size cells. But the number of cells differed between monkeys (1700 vs 2249) because of small differences in chamber placement and cortical anatomy.

Line 313: A distances metric is defined to be zeros to itself, dist(A,A) = 0. The correlation of A with itself is 1. Do you mean to say that you used 1-max_correlation, or that you used the correlation as a similarity metric?

This inquiry pertains to the clustering analysis, which has been removed from the manuscript. Nevertheless, to address the reviewer’s question, correlation was indeed used as a similarity metric.

Line 323: My guess the point of this analysis is that the total distances decrease with an increasing number of clusters (and the correlation increases with an increasing number of clusters). So, fitting a line assumes a linear relationship between these two variables. When you say "The longest of those orthogonal lines identified the optimal number of clusters" why are the lines orthogonal? By longest – do you mean the largest deviation between the linear fit either above or below the line? This would pick either an especially good or bad parcellation. Or did you look at deviations above or below only?

This inquiry pertains to the clustering analysis, which has been removed from the manuscript. Nevertheless, to address the reviewer’s question, the orthogonal lines were always below the fitted line. This is a function of how the line was fitted to the graph in the first place. Thus, criteria were consistent for selecting clusters.

Line 327: In the methods, we have 3 conditions – in the results 4. Which one is correct?

The task involved 4 total conditions: 3 movement conditions and 1 withhold condition. We have checked the entire manuscript and revised any inconsistencies.

Line 370: Is the darkening appearing at 1s an artifact of the measurement or real? At least for the posterior border, it looks like it follows exactly the border of the window.

The posterior border of the field-of-view coincides with the central sulcus, which contains a major blood vessel. The darkening is most likely due to reflectance change in vessels as opposed to an edge effect of the cranial window. We revised the text to indicate this point (line 157-162)

Line 451: "that shifted and expanded across the time series" – is there a statistical test for the claim that the activity region shifted?

We revised the statement to convey that it is a qualitative description of the spatio-temporal pattern of cortical activity (line 254-255).

Line 462: To what degree is the fact the activity averaged over the entire window remains stable simply a consequence of the high-pass filtering applied to each image? High-pass filtering basically removes the mean across the entire window, no?

We tested the impact of the high-pass filter on the reflectance change time courses from a region-of-interest that covers the entire field of view. Thus, we systematically varied the size of the filtering kernel (100-800 pixels in steps of 100 pixels) then examined the time courses. The results of those tests are plotted in Author response image 1 for the precision grip (movement condition) and withhold conditions (no movement). We note that in both conditions, most of the colored lines overlapped. This indicates that kernel size had little/no impact on time courses measured from the large ROIs. Nevertheless, the necessity for high pass filtering is evident from the gray time course, which was generated without high pass filtering. That line plot shows the dominance of global signals over the intrinsic signals of interest.

**Author response image 1. sa2fig1:** 

Line 466: What is meant by "domains"? Activity clusters?

Yes, domains are activity clusters. The term was defined on line 357 in the original submission (now line 145-149).

Line 5E: The analysis here involves statistical testing within each session, and then averaging activity estimates across sessions, subject to an arbitrary 50% threshold for statistical significance within the session. I cannot see a good motivation for this awkward type of analysis. If you want to consider trials a random effect, I would recommend calculating a statistical test for all trials across sessions combined. If you want to consider sessions as a random effect, then calculate a t-test across the 8 repeated activity estimates per session – this analysis also automatically takes the variability across trials into account. Or is there another reason why you chose this baroque style of analysis that is mentioned?

We agree with the reviewer and have overhauled the analysis. We now pool trials across sessions then conduct a t-test. Thus, for every condition, only one t-test is calculated (Figure 5G and 5J).

Line 474: "organization of domains was more similar across conditions within an animal than for the same condition across animals". See Ejaz et al. (2015). Nature Neuroscience, for a similar observation in human fMRI motor maps.

We agree and now cite Ejaz et al. (2015) on line 315.

Line 483: The overlap analysis is unfortunately quite dependent on the threshold.

overhauled our thresholding procedure so that it is now based on a t-test that includes all trials. We believe that the thresholded maps are now considerably more robust than in the original submission. We provide more detail in our response to the first major point from the same reviewer.

Line 494: form → from?

Revised.

I am not sure what I am supposed to learn from the cluster analysis. It is entitled "Clustering time courses recapitulates spatial patterns of activity" – likely referring to the similarity between Figure 6a and 6b. As the clustering partly depends on the magnitude of the signal at the time points presented in Figure 6a, what is the alternative? Is this similarity more than expected by chance?

The clustering analysis has been removed from the manuscript.

Nevertheless, yes, similarities between Figure 6A and 6B drove our conclusion that clustering returned maps that were comparable to the activity maps from averaging frames. It is important to note that clustering was conducted on the 70 time points that make up a time series (Figure 6B and 6E in original submission), whereas activity maps were averages from just five time points (Figure 6A and 6D in original submission).

Indeed, we believe that the similarities were more than expected by chance. We estimated the noise ceiling from running the clustering analysis after shuffling the timecourse of every pixel. This perturbation changed the temporal profile of every pixel but preserved signal magnitude and spatial relationships between pixels. Clustering the shuffled time courses returned salt and pepper maps that had no resemblance to original Figure 6A-E. We should have included this control in the original submission.

Line 527-546. The overlap between the maps is very hard to interpret, as it is highly dependent on the threshold procedure applied, which is somewhat arbitrarily chosen (see above). Even random maps of a certain size would overlap to some degree, and even identical "true" maps don't overlap completely due to measurement error. An overlap of 0.5 has no intrinsic meaning. It would probably be more informative to report the correlation between unthresholded maps, and compare this to a noise ceiling, where you correlate the maps of one condition across sessions.

We agree and have overhauled the analysis in question. Conditions are now compared in two ways. First, direct statistical comparison between the reach-to-grasp condition and the reach-only condition (Figure 6B-C). Second, correlating average time series from pairs of conditions (Figure 6D). Both comparisons reported differences between the precision grip condition and the reach-only condition. Those two conditions had the largest difference in forelimb joint kinematics and muscle activity (Figure 7).

Reviewer #2 (Recommendations for the authors):Chehade and Gharbawie investigated motor and premotor cortex in macaque monkeys performing grasping and reaching tasks. They used intrinsic signal optical imaging (ISOI) covering an exceedingly large field-of-view extending from the IPS to the PS. They compared reaching and fine/power-grip grasping ISOI maps with "motor" maps which they obtained using extensive intracranial microstimulation. The grasping/reaching-induced activity activated relatively isolated portions of M1 and PMd, and did not cover the entire ICM-induced 'motor' maps of the upper limbs. The authors suggest that small subzones exist in M1 and PMd that are preferentially activated by different types of forelimb actions. In general, the authors address an important topic. The results are not only highly relevant for increasing our basic understanding of the functional architecture of the motor-premotor cortex and how it represents different types of forelimb actions, but also for the development of brain-machine interfaces. These are challenging experiments to perform and add to the existing yet complementary electrophysiology, fMRI, and optical imaging experiments that have been performed on this topic – due to the high sensitivity and large coverage of the particular IOSI methods employed by the authors. The manuscript is generally well written and the analyses seem overall adequate – but see below for some additional analyses that should be done. Although I'm generally enthusiastic about this manuscript, there are two major issues that should be clarified. These major questions relate mainly to potential thresholding issues and clustering issues.1) The main claim of the authors is that specific forelimb actions activate only a small fraction of what they call the motor map (i.e., those parts of M1/PMd that evoke muscle contractions upon ICM). The action-related activity is measured by ISOI. When looking a the 'raw' reflectance maps, it is rather clear that relatively wide portions of the exposed cortex are activated by grasping/reaching, especially at later time points after the action. In fact, another reading of the results may be that there are two zones of 'deactivation' that split a large swath of motor-premotor cortex being activated by the grasping/reaching actions. (e.g. at 6 seconds after the cue in Figure 3A, 5A). At first sight, the 'deactivated' regions seem to be located in the cortex representing the trunk/shoulder/face – hence regions not necessarily activated (or only weakly) during the grasping/reaching actions. If true, this means that most of the relevant M1/PMd cortex IS activated during the latter actions – opposing the 'clustering' claims of the authors. This raises the question of whether the 'granularity' claimed by the authors is: a. Threshold dependent. In this context, the authors should provide an analysis whereby 'granularity' is shown independent of statistical thresholds of the ISOI maps.

We appreciate the reviewer’s concerns and have completely revised the analyses central to Figure 5. We believe that the figure now contains evidence from both thresholded and unthresholded ISOI data in support of limited spatial extent of cortical activation (i.e., “granularity” in the reviewer’s comments).

For evidence from unthresholded ISOI data, we examined reflectance change time courses from different size ROIs (line 764-768). (A) Small circular ROIs (0.4 mm radius), which we placed in the M1 hand, M1 arm, and PMd arm, zones (Figure 5B). (B) Large ROI inclusive of the M1 and PMd forelimb representations (Figure 5B). We reasoned that if cortical activity is spatially widespread, then the small and large ROIs would report similar time courses. In contrast, if cortical activity is spatially focal, then activity would be detected in the small ROI time courses but would washed out in the large ROI time courses. Our results support the second possibility (Figure 5C-F). Thus, in the movement conditions, time courses from the small ROIs had a large negative peak after movement completion (Figure C-E). In contrast, the characteristic negative peak was absent in the time courses obtained from the large ROI (Figure 5F).

Separately, we revised our thresholding approach to make those results less sensitive to thresholding effects (more details in our response to the first major point from Reviewer 1). The revised results – thresholded/ binarized maps – are consistent with focal cortical activity. Figure 5G & 5J show activity maps thresholded (t-test, p<0.0001) without correction for multiple comparisons, and therefore represent the least restrictive estimate of the spatial extent of cortical activity. Measurements from these maps showed that significantly active pixels overlapped <40% of the M1 & PMd forelimb representations. We interpret the thresholded results as evidence in support of focal cortical activity.

b. Dependent on the time-point one assesses the maps. Given the sluggish hemodynamic responses, it is unclear which part of the ISOI maps conveys the most information relative to the cue and arm/hand movements. I suspect that timepoints > 6 s will reveal even larger 'homogeneous' activations compared to the maps < 6s.

We agree with the reviewer that the lag in hemodynamic signals complicates frame selection. Nevertheless, it is unlikely that cortical activity maps would have been larger at time points >6s from Cue. We provide three supporting arguments.

(1) In the imaging sessions used in Figure 4, we acquired images for 9s per trial and systematically varied Cue onset time. The time courses in Figure 4A-B show that for all Cue onset conditions, the negative peak occurred <6s from Cue. This observation from unthresholded results does not support the notion of greater cortical activity at time points >6s from Cue.

(2) From the same experiment, Figure 4C shows 9 thresholded/binarized maps generated from different time points in relation to Cue. We measured the size of each map (i.e., overlap with the M1/PMd forelimb representations). We present the results in a scatter plot in Author response image 2. The largest maps came from an average frame captured +5.8-6.0s from Cue. Those maps are on the diagonal in Figure 4E (top left to bottom right). This result from thresholded data therefore does not support the notion of greater cortical activity at time points >6s from Cue.

(3) In all other sessions, we acquired images for 7s per trial (-1.0 to +6.0 s from Cue) without varying Cue onset time. At every time point (100 ms), we measured the size of the thresholded/binarized map in relation to the size of the M1 and PMd forelimb representations. The results are presented in Figure 5I & 5L and indicate that thresholded maps plateau in size by 5.0-5.5 s from Cue. At peak size, the maps overlapped <50% of the M1 and PMd forelimb representations. These result indicates that it is unlikely that we underreported the size of activity maps by not measuring map size beyond 6s from Cue.

In fact, Figure 5F (which is highly thresholded) shows a surprisingly good match between the different forelimb actions, which argues against the existence of small subzones that are preferentially activated by different types of forelimb actions -the main claim of the authors.

Our original proposal should have been more clearly stated. We were proposing that the thresholded maps, which had similar spatial organizations across conditions as the reviewer suggested, reported on subzones tuned for reach-to-grasp actions. Adjacent to those subzones could be other subzones that are preferentially active during other types of forelimb actions (e.g., pulling, pushing, grooming). We could not test this possibility in our study because the behavioral task examined a narrow range of arm and hand actions. We therefore revised the Discussion to state the limitations of our task and to lean more on published work that supports the present proposal (439-443 and 504-508).

2) Related to the previous point, the ROI selections/definitions for the time course analyses seem highly arbitrary. As indicated in the introduction, the clustering hypothesis dictates that "an arm function would be concentrated in subzones of the motor arm zones. Neural activity in adjacent subzones would be tuned for other arm functions." To test this hypothesis directly in a straightforward manner, the authors could use the results from the ICM experiment to construct independent ROIs and to evaluate the ISOI responses for the different actions. In that case, the authors could do a straightforward ANOVA (if the data permits parametric analyses) with ROI, action, and time point (and possibly subject) as factors.

We agree with the reviewer, and we now leverage the ICMS map for guiding ROI placement. All time courses are now derived from 1 of 2 types of ROIs. (1) Small ROIs (0.4 mm radius) placed in zones defined from ICMS (e.g., M1 hand zone). (2) Large ROIs that include the entire forelimb representations in M1 or in PMd (Figure 5B).

3) More details about the transparent silicone membrane should be provided: How implanted? How maintained? Please provide pictures at the end of the 9 and 22months periods.

We expanded the Methods section to provide additional information on the silicone membrane (line 566-577).

The optical imaging chamber and the silicone membrane were not implanted until a monkey was considered trained on the task (9 and 22 months, respectively). We revised the Methods to clarify this point (line 566-567).

4) Figure 1D: Are the yellow circles task cues that the monkeys saw? If not, what were the different cues (Reach, Grasp, Withhold)? Were precision and power grip (and reach?) trials similarly cued?

The yellow circles were digitally added to the still frames to make the target obvious to the reader. The monkey did not see the yellow circles. We revised the figure legend to clarify this point (line 1005).

Identical Go Cues were provided for the 3 movement conditions (i.e., reach-to-grasp large sphere, reach-to-grasp small sphere, and reach only). The purpose of the Go Cue was to instruct the monkey to start reaching. In any given trial, the target object was visible to the monkey throughout the trial. Thus, the type of movement condition was not cued as it was self-evident at the outset of the trial. We revised the Methods to ensure that there is no confusion about Cues (line 583-585; 593-594; 611-612).

5) For the MEG activity a total of 14 27gauge wires were inserted in 7 muscles. This sounds rather invasive. Did monkeys tolerate this easily? Were local anesthetics used? How deep were the wires inserted in the muscle? How did one prevent the monkeys grabbing these wires?

EMG recordings were conducted in separate sessions from ISOI. Once the monkey was head-fixed, we administered a single, low dose of ketamine (2-3 mg/kg, IM). The point of this mild sedation was to minimize any discomfort for the animal during EMG wire insertion, and to create a safe experience for the animal and the handler. The sedation achieved both objectives without eliminating reflexes or muscle tone. We have no reason to believe that the monkey did not readily tolerate the wire insertion procedure or the EMG recordings. Behavioral testing and EMG recordings started 45-60 minutes after sedation. By that point, there were no outwardly lingering effects of sedation and the monkey rapidly reached and grasped treats from the experimenter (line 633-643). Also, there were no differences in reaction time, reach speed, or success rate, on sessions without mild sedation and sessions with mild sedation and EMG recordings.

EMG wires were inserted ~15 mm below the surface of the skin.

Only the non-working forelimb could have disturbed the EMG wires that we inserted into the working forelimb. But the non-working forelimb was restrained from the outset. This point is now in the Methods (line 642-643).

6) The large FOV was illuminated using 2-3 LEDs. Was illumination uniform throughout the FOV? If not, can this lead to inhomogeneous sensitivity?

Even illumination is essential for high quality ISOI. We invested effort at the start of every session to optimize illumination. We revised the Methods (line 730-732) to include two factors that were critical to achieving even illumination. (1) Diffuser lens attached to each LED. (2) Real-time heat map of the field-of-view for positioning the LEDs and adjusting the intensity of each one.

7) Pixel values were clipped, if I understood correctly, at either 0.3 SD or 1 SD from the mean. This significantly changes the dynamic range of these pixel values. Are results different without clipping?

Pixel values were clipped only to facilitate visualization of optical images (e.g., Figure 3A, 5A, 6A). No clipping was done wherever quantitative measurements were involved (e.g., Figure 5G-L). Thus, there was no clipping for time courses, t-tests, and spatial measurements. We revised the Methods to clarify this point (line 759-761).

8) An arbitrarily p<0.0001 statistical threshold was used. Why was no correction for multiple correspondence performed as one collects data from ~ 600k-1400k pixels (yet without considering the masks)?

We redid the analyses after Bonferroni correction for multiple comparisons (p<0.1e-7; 0.05/350k pixels [excluding masked pixels]). We describe the correction in Methods (line 791-794). We now report results thresholded at p<1e-4 and thresholded at p<1e-7 (Figure 5G-L; Figure 5 – Supplemental Figure 1). Not surprisingly, maps were smaller in the more conservative threshold.

We note that many labs consider the Bonferroni correction an overly conservative approach for correcting family wise errors in imaging. The correction assumes that pixels (or voxels in fMRI) provide independent measurements, which is not the case given the close functional relationship between every pixel with its neighbors. An alternative approach would have been to determine the number of multiple comparisons from the number of ROIs. But we had only 4 ROIs (Figure 5B), which would have resulted in a threshold (p<0.01, [p=0.05/4]) that is more lenient than the one adopted here and in the original submission (p<0.0001).

9) What is the reason for obvious edge effects in the withhold condition (Figure 3A, 3F) and apparently also the precision grip condition (Figure 5A: 1.1-3.5sec)? Is this artifactual?

We understand the reviewer’s concerns, which were also raised by the first reviewer. We believe that the seeming edge effect is activity in major blood vessels. Specifically, the posterior border of the field-of-view coincides with the central sulcus, which contains a major blood vessel. The darkening is most likely due to reflectance change in the vessel as opposed to an edge effect of the cranial window. Similarly, the rostral edge effect coincides with the arcuate sulcus. We now include this explanation in the Results (line 157-160).

10) How are the ROIs defined (cfr major point 2)? They differ in size and location between analyses (e.g. Figure 3 vs Figure 5/S1). Please discuss.

We agree with the reviewer that the ROIs differed in size and location across analyses. We now use 1 of 2 types of ROIs. (1) Circular ROI that is the same size across analyses. (2) ROI fitted to the forelimb representation in M1 or PMd. The shape and size of those ROIs is guided exclusively by the motor map. We revised the Methods to reflect these points (line 764-768).

11) I find it a bit counter-intuitive that the same color-code (i.e. black values) is given for 'activations' in panel 3G and 'deactivations' in panel 3H. Why not using red and blue instead? Along the same vein: activations in the time-courses are negative (reflecting the darkening). It would be instructive for the non-experienced reader to either add activation/ suppression on the figures (above and below zero in Figures 3I and J and 5C), or to invert the Y-axis.

We agree with the reviewer on all accounts. We revised the panels in Figure 3 so that significant pixel darkening is shown in red and significant pixel brightening is shown in blue. Separately, we revised the y-axis labels for all time course plots to facilitate interpretation of negative and positive values.

12) Line 426: Figure 5K does not exist.

We corrected the error.

13) Line 520: It is argued that the unsupervised cluster analysis, which is quite interesting, is similar in both monkeys. However, this is not obvious from the data: neither in the spatial domain (Figure 6B vs S2B) nor in the temporal domain Figure 6C and S2 C, especially the blue plots. In fact, the clustering data from monkey G reveal a rather widespread, uninterrupted pattern arguing against the 'cluster hypothesis' of the authors. This should be discussed in more depth.

We agree with the reviewer’s concerns. As stated in response to reviewer 1, we replaced the clustering analysis with a more straightforward analysis that we now present in Figure 6 (line 345-374).

14) Figure 7A: It is unclear which analysis was done. Is this simply giving 3 different colors to each pixel – indicating 1) precision > baseline; 2) reach > baseline 3) both > baseline (for the left panel)? Why not performing straight (pair-wise) subtractions?

We agree that the analysis and the presentation of the results were both confusing. We replaced the analysis with pairwise comparisons using t-tests and cross-correlations (Figure 6, line 345-374).